# Predicting referral need for febrile children in low-resource community settings in South and Southeast Asia

In resource-constrained community settings, identifying which febrile children require referral remains a major unmet need. Current World Health Organization (WHO) danger signs have limited accuracy, resulting in missed severe illness and unnecessary referrals. Here we developed and validated clinical prediction models to support referral decisions using data from 3,405 children aged 1–59 months presenting with community-acquired acute febrile illnesses to seven hospitals across Bangladesh, Cambodia, Indonesia, Laos and Vietnam. Cambodian data were held out for external validation. The model using simple clinical parameters (sensitivity 74.7% (95% confidence interval (CI): 59.4–88.1); specificity 99.1% (95% CI: 97.7–99.7)) outperformed WHO criteria (sensitivity 55.5% (95% CI: 39.4–72.7); specificity 82.6% (95% CI: 77.1–87.6)) for identification of children at risk of severe disease (death or organ support within 2 days). Including either pulse oximetry or the host biomarker soluble TREM1 (sTREM1) increased sensitivity to 88.9% (95% CI: 76.7–97.8; pulse oximetry) and 89.2% (95% CI: 76.9–97.5; sTREM1), respectively. The pulse oximetry-based model achieved these gains with a threefold reduction in referral rates. These approaches appear cost-effective (pulse oximetry incremental cost effectiveness ratio (ICER) = $26.28; sTREM1 ICER = $196.46) and could improve triage for febrile illness in low-resource settings by enabling more accurate referral decisions. They warrant evaluation in community-based trials.

Infectious diseases account for the majority of the 2.5 million deaths that occur each year among children aged 1–59 months[1]. Many deaths happen in community settings, because a child does not attend health services, a child presents too late, there are financial or logistical barriers to referral or there is failure to recognize impending critical illness[2–4]. Early identification of children at risk of life-threatening infection remains difficult[5,6]. Most Early Warning Scores (EWSs) rely on combinations of abnormal vital signs (for example, fever, tachycardia and tachypnea), but these are dynamic, prone to confounding and poorly sensitive and specific for identifying early stages of sepsis[7,8].

In resource-constrained community settings, current WHO Integrated Management of Childhood Illnesses (IMCI) guidelines do not advocate the use of vital signs to assess illness severity[9,10]. Instead, hospital referral is prompted by clinical danger signs, such as convulsions, intractable vomiting, lethargy or prostration. The accuracy of these indicators is suboptimal, and they are prone to considerable interobserver variability[7,11–13]. Neither vital signs nor danger signs reliably stratify risk in common childhood infections, underscoring the need for better prognostic tools[14]. Such tools would be particularly valuable in remote and conflict-affected settings, where capacity for safety-netting is typically minimal while the opportunity cost of referral is high.

One option, spotlighted by the COVID-19 pandemic, is expansion of pulse oximetry to primary care settings. Championed as the

✉e-mail: arjun@tropmedres.ac

'fifth vital sign'[15], hypoxemia predicts poor outcomes in childhood illnesses, including pneumonia, meningitis, malaria and malnutrition, and identifies large numbers of children requiring hospital referral who are missed by IMCI guidelines alone[16-20]. Despite this promise, pulse oximetry remains underutilized in community settings. Barriers include availability, cost and usability of appropriately sized probes, accuracy across skin tones and staff capacity and workload; oximetry also takes time, especially in young children where measurement can be challenged by movement or crying[21-23].

Another strategy is integration of clinical assessment with host biomarker testing[5,24]. Circulating markers of immune and endothelial activation predict disease severity agnostic to pathogen etiology and, in certain contexts, have outperformed pulse oximetry[25-29]. We recently reported promising prognostic performance of individual host biomarkers at the community level[30], and an operational evaluation among village malaria workers in Cambodia demonstrated the feasibility of this approach[31]. If effective, such strategies could capitalize on point-of-care testing capacity developed within community healthcare worker networks over preceding decades by control programs for diseases such as malaria and HIV[32].

Febrile illness refers to the presence of fever (typically ≥38 °C)[9,10], a common clinical feature of many childhood infections. Although most febrile episodes are self-limiting and caused by viral, bacterial or parasitic pathogens, individual clinical features rarely indicate which children will progress to severe disease. Building upon our previous work and using the same multicountry dataset[30], our objective in this paper was to address a complementary question: how prognostic information can be integrated into risk prediction models to guide referral decisions for febrile children in resource-constrained community contexts and decentralized models of care. Whereas our previous analysis focused on the predictive performance of individual host biomarkers for disease progression, the present study develops and validates multivariable clinical prediction models comprising simple clinical parameters (for example, vital signs and danger signs) and evaluates the added utility of including pulse oximetry and host biomarker testing for referral decision-making.

## Results

### Study population

A total of 11,962 children were screened between 5 March 2020 and 4 November 2022. Of these, 3,998 were eligible (3,998/11,962, 33.4%) and 3,423 were recruited (575/3,998, 14.4% refusal rate). There were two withdrawals and 16 children with incomplete follow-up data, leaving 3,405 participants for further analyses (Extended Data Fig. 1).

Median age was 16.8 months (interquartile range (IQR), 8.7–31.0), and 59.6% (2,029/3,405) of the cohort were male. Malnutrition was prevalent: 17.2% (585/3,393) of children were wasted (weight-for-height $z$-score < −2), and 19.5% (664/3,401) were stunted (height-for-age $z$-score < −2). Of the 585 children with Global Acute Malnutrition, 248 (42.4%) had Severe Acute Malnutrition (weight-for-height $z$-score < −3). Median symptom duration prior to presentation was 3 days (IQR, 2–4). Acute respiratory infections were the most common reason for presentation, followed by diarrheal syndromes and undifferentiated febrile illnesses. A quarter of participants had a microbiological cause for their infection identified (898/3,405, 26.4%; Supplementary Table 1). Most children lived close to the hospital (2,777/3,405, 81.6% within 1 hour). In total, 1,342 participants (1,342/3,405, 39.4%) had received care in the community at an earlier point in their illness; 193 (193/3,405, 5.7%) had received parenteral treatment, and none had been admitted.

Baseline characteristics were largely balanced across derivation and validation cohorts (Table 1), although the validation cohort was younger (median age, 13.1 months versus 18.7 months), and a higher proportion had received parenteral treatment in the community at an earlier point in their illness (109/824, 13.2% versus 84/2,581, 3.3%). Baseline biomarker concentrations were similar across the cohorts.

There was a greater proportion of confirmed viral infections in the derivation cohort (709/2,581, 27.4% versus 155/824, 18.7%). Bacteremia rates were similar (12/781, 1.5% versus 7/411, 1.7%), although blood cultures were collected more frequently from inpatients in the validation cohort (411/644, 63.8% versus 781/2,080, 37.5%).

### Progression to severe febrile illness

Overall, 133 children met the primary outcome (defined as death or organ support within 2 days of enrollment; 133/3,405, 3.9%): 111 survivors who required organ support (non-invasive ventilation, $n = 59$; mechanical ventilation, $n = 43$; inotropic therapy, $n = 19$) and 22 deaths (derivation cohort = 97/2,581, 3.8% and validation cohort = 36/824, 4.4%). Weighted outcome prevalence was 0.34% (95% CI: 0.28−0.41), which was similar across derivation (0.36%, 95% CI: 0.29−0.45) and validation (0.30%, 95% CI: 0.20−0.42) cohorts.

Among the candidate clinical predictors (Supplementary Table 2), association with the outcome was similar across derivation and validation cohorts (Table 1), although a weaker association was observed for altered mental state, prolonged capillary refill time and presence of WHO danger signs in the validation cohort. Most candidate biomarker predictors were associated with the outcome across derivation and validation cohorts, apart from C-reactive protein (CRP), IL-1ra, IL-6, IL-10, glucose and hemoglobin, which were not associated with the outcome in the validation cohort.

### Clinical prediction models

When each of the 17 biomarkers was included in turn alongside the candidate clinical predictors, 11 were retained after backward stepwise selection in the derivation dataset. Similar results were obtained with and without the inclusion of peripheral oxygen saturation ($SpO_2$) as a candidate predictor (Supplementary Tables 3 and 4). In a sensitivity analysis excluding the northern Vietnam site (Methods and Supplementary Table 5), a biomarker was retained in seven of the models. Across all analyses, the clinical biomarker models containing sTREM1 consistently demonstrated the highest discrimination. Thus, four models were taken forward for validation: the clinical model (clinical parameters only), the pulse oximetry model (clinical parameters plus $SpO_2$), the sTREM1 model (clinical parameters plus sTREM1) and the combined model (clinical parameters plus $SpO_2$ and sTREM1).

Heart rate, respiratory rate, prostration and intractable vomiting were consistently selected across all models ($n = 4$); altered mental state was selected in two models; and prolonged capillary refill time and convulsions were selected in one model each (Table 2). Model equations and odds ratios are reported in the appendix (Supplementary Table 6).

Discrimination and calibration of the models in the validation cohort are presented (Table 2). Discrimination ranged from a weighted area under the receiver operating characteristic curve (AUC) of 0.95−0.98. Calibration was best at lower predicted probabilities, with some underestimation of risk in individuals with the highest predicted probabilities of severe disease.

### Clinical utility

Clinical utility of the models for triage of febrile children was assessed within a hypothetical traffic light framework whereby any child with a predicted probability of developing severe disease lower than 0.5% (the rule-out threshold) is discharged (green); those with a predicted probability higher than 2% (the rule-in threshold) are referred for higher-level care (red); and those with predicted probabilities between 0.5% and 2% are monitored (amber)−for example, at the rural clinic or via telephone or outreach follow-up (Fig. 1).

Within this framework, using the clinical model, one in 10 children recommended for referral would have developed severe disease. However, the clinical model would have failed to identify 25% of children who progressed to severe disease, in whom discharge would have been recommended (sensitivity 74.7% (95% CI: 59.4−88.1)). Compared to the

**Table 1 | Baseline characteristics of the derivation and validation cohorts, stratified by whether a child progressed to develop severe febrile illness**

| Characteristic | Derivation cohort | | | | Validation cohort | | | |
|---|---|---|---|---|---|---|---|---|
| | Overall n=2,581 | Non-severe n=2,484 | Severe n=97 | Pvalue | Overall n=824 | Non-severe n=788 | Severe n=36 | Pvalue |
| Demographics and background | | | | | | | | |
| Age (months) | 18.7 (9.3, 32.9) | 18.9 (9.5, 33.1) | 6.9 (3.2, 25.8) | <0.0001 | 13.1 (7.5, 24.4) | 13.5 (8.1, 24.9) | 3.4 (1.8, 6.2) | <0.0001 |
| Male sex | 1,577 (61%) | 1,513 (61%) | 64 (66%) | 0.32 | 452 (55%) | 428 (54%) | 24 (67%) | 0.15 |
| Known comorbidity | 65 (2.5%) | 63 (2.5%) | 2 (2.1%) | 1.00 | 37 (4.5%) | 37 (4.7%) | 0 (0%) | 0.40 |
| Recent hospitalization[a,*] | 277 (11%) | 265 (11%) | 12 (13%) | 0.58 | 152 (19%) | 147 (19%) | 5 (14%) | 0.46 |
| Anthropometrics | | | | | | | | |
| Wasted (WHZ < −2)[*] | 474 (18%) | 445 (18%) | 29 (30%) | 0.0030 | 111 (14%) | 106 (14%) | 5 (14%) | 0.80 |
| Stunted (HAZ < −2)[*] | 484 (19%) | 462 (19%) | 22 (23%) | 0.31 | 180 (22%) | 167 (21%) | 13 (36%) | 0.035 |
| MUAC-for-age z-score[b,*] | −0.2 (−1.1, 0.7) | −0.2 (−1.1, 0.7) | −0.5 (−1.4, 0.7) | 0.12 | −0.5 (−1.2, 0.2) | −0.5 (−1.1, 0.2) | −0.8 (−1.7, 0.3) | 0.40 |
| Illness history | | | | | | | | |
| Duration of illness (days) | 3.0 (2.0, 4.0) | 3.0 (2.0, 4.0) | 3.0 (2.0, 5.0) | 0.020 | 3.0 (2.0, 4.0) | 3.0 (2.0, 4.0) | 3.5 (3.0, 5.0) | 0.051 |
| Care prior to presentation[c] | 864 (33%) | 829 (33%) | 35 (36%) | 0.58 | 478 (58%) | 457 (58%) | 21 (58%) | 0.97 |
| Travel time ≤1 hour[d] | 2,227 (86%) | 2,150 (87%) | 77 (79%) | 0.044 | 550 (67%) | 532 (68%) | 18 (50%) | 0.029 |
| Presenting syndrome[e] | | | | | | | | |
| URTI | 795 (31%) | 773 (31%) | 22 (23%) | 0.077 | 326 (40%) | 309 (39%) | 17 (47%) | 0.34 |
| LRTI | 930 (36%) | 877 (35%) | 53 (55%) | 0.0001 | 417 (51%) | 384 (49%) | 33 (92%) | <0.0001 |
| Diarrheal | 397 (15%) | 386 (16%) | 11 (11%) | 0.26 | 249 (30%) | 245 (31%) | 4 (11%) | 0.011 |
| No focus | 372 (14%) | 366 (15%) | 6 (6.2%) | 0.019 | 155 (19%) | 153 (19%) | 2 (5.6%) | 0.037 |
| Neurological | 372 (14%) | 359 (14%) | 13 (13%) | 0.77 | 58 (7.0%) | 57 (7.2%) | 1 (2.8%) | 0.51 |
| WHO danger signs | | | | | | | | |
| Any danger sign present[*] | 1,246 (48%) | 1,171 (47%) | 75 (78%) | <0.0001 | 361 (44%) | 341 (43%) | 20 (56%) | 0.15 |
| Prostration | 199 (7.7%) | 155 (6.2%) | 44 (45%) | <0.0001 | 41 (5.0%) | 37 (4.7%) | 4 (11%) | 0.098 |
| Intractable vomiting[*] | 604 (23%) | 564 (23%) | 40 (41%) | <0.0001 | 83 (10%) | 81 (10%) | 2 (5.6%) | 0.57 |
| Convulsions[*] | 373 (14%) | 360 (15%) | 13 (14%) | 0.79 | 60 (7.3%) | 59 (7.5%) | 1 (2.8%) | 0.51 |
| Lethargy[*] | 520 (20%) | 478 (19%) | 42 (44%) | <0.0001 | 305 (37%) | 286 (36%) | 19 (53%) | 0.045 |
| Vital signs | | | | | | | | |
| Heart rate (bpm)[*] | | | | | | | | |
| 1–12 months | 154.0 (140.0, 171.0) | 153.0 (140.0, 170.0) | 170.0 (152.0, 184.0) | <0.0001 | 159.0 (143.0, 176.0) | 157.0 (142.0, 173.0) | 177.0 (161.0, 186.0) | <0.0001 |
| 12–60 months | 140.0 (126.0, 157.0) | 140.0 (126.0, 156.0) | 159.0 (140.0, 179.5) | <0.0001 | 140.0 (128.0, 162.0) | 140.0 (128.0, 162.0) | 167.0 (146.0, 172.0) | 0.17 |
| Respiratory rate (bpm)[*] | | | | | | | | |
| 1–12 months | 45.0 (38.0, 55.0) | 44.0 (38.0, 53.0) | 60.0 (50.0, 68.0) | <0.0001 | 44.0 (37.0, 54.0) | 44.0 (36.0, 52.0) | 58.0 (46.0, 64.0) | <0.0001 |
| 12–60 months | 35.0 (30.0, 41.0) | 35.0 (30.0, 41.0) | 39.0 (31.5, 55.0) | 0.016 | 38.0 (32.0, 45.0) | 38.0 (32.0, 45.0) | 54.0 (50.0, 67.0) | 0.032 |
| SpO₂ in room air (%)[*] | 98.0 (97.0, 99.0) | 98.0 (97.0, 99.0) | 97.0 (95.0, 98.0) | 0.0004 | 99.0 (98.0, 100.0) | 99.0 (98.0, 100.0) | 96.0 (94.5, 97.5) | <0.0001 |
| Axillary temperature (°C)[*] | 37.7 (37.0, 38.4) | 37.7 (37.0, 38.4) | 38.0 (37.2, 38.6) | 0.098 | 37.3 (36.7, 38.1) | 37.3 (36.7, 38.1) | 37.0 (36.5, 37.5) | 0.019 |
| CRT > 2 seconds | 180 (7.0%) | 153 (6.2%) | 27 (28%) | <0.0001 | 14 (1.7%) | 12 (1.5%) | 2 (5.6%) | 0.12 |
| Not alert[f] | 58 (2.2%) | 41 (1.7%) | 17 (18%) | <0.0001 | 28 (3.4%) | 26 (3.3%) | 2 (5.6%) | 0.35 |
| Endothelial activation markers | | | | | | | | |
| ANG-1 (pg ml⁻¹)[*] | 6,396.5 (3,387.5, 11,645.0) | 6,411.0 (3,387.0, 11,607.0) | 6,049.0 (3,426.0, 13,382.0) | 1.00 | 5,686.5 (3,454.0, 9,139.0) | 5,713.5 (3,454.0, 9,240.0) | 4,621.0 (3,440.5, 7,421.5) | 0.24 |
| ANG-2 (pg ml⁻¹)[*] | 1,330.0 (998.5, 1,844.0) | 1,320.0 (993.0, 1,802.0) | 2,343.0 (1,317.0, 4,608.0) | <0.0001 | 1,455.5 (1,090.0, 2,051.0) | 1,427.0 (1,081.0, 1,965.0) | 3,008.0 (2,119.0, 4,254.5) | <0.0001 |
| sFLT-1 (pg ml⁻¹)[*] | 183.0 (149.0, 231.0) | 181.0 (148.0, 227.0) | 258.0 (196.0, 369.0) | <0.0001 | 196.0 (156.0, 245.0) | 194.5 (154.0, 242.0) | 260.5 (227.0, 415.5) | <0.0001 |
| Immune activation markers | | | | | | | | |
| CHI3L1 (ng ml⁻¹)[*] | 23.8 (15.7, 37.7) | 23.5 (15.5, 36.9) | 37.6 (22.8, 62.6) | <0.0001 | 28.9 (20.2, 43.1) | 28.5 (20.1, 42.5) | 38.2 (26.9, 72.8) | 0.0037 |
| CRP (mg l⁻¹)[*] | 13.2 (3.2, 49.3) | 12.8 (3.2, 47.8) | 31.4 (6.8, 150.5) | 0.0001 | 17.3 (4.8, 55.3) | 17.5 (4.8, 55.3) | 11.8 (2.1, 55.5) | 0.31 |
| IL-1ra (pg ml⁻¹)[*] | 2,036.0 (980.0, 4,875.0) | 1,983.0 (967.0, 4,764.0) | 4,117.0 (1,756.0, 14,784.0) | <0.0001 | 1,755.0 (789.0, 4,429.0) | 1,781.0 (806.0, 4,440.0) | 1,505.0 (728.5, 3,509.5) | 0.32 |

**Table 1 (continued) | Baseline characteristics of the derivation and validation cohorts, stratified by whether a child progressed to develop severe febrile illness**

| Characteristic | Derivation cohort | | | | Validation cohort | | | |
|---|---|---|---|---|---|---|---|---|
| | Overall n=2,581 | Non-severe n=2,484 | Severe n=97 | P value | Overall n=824 | Non-severe n=788 | Severe n=36 | P value |
| IL-6 (pg ml⁻¹)* | 18.4 (7.7, 46.3) | 17.6 (7.6, 43.5) | 73.7 (20.9, 320.0) | <0.0001 | 16.9 (7.2, 46.9) | 16.8 (7.2, 46.9) | 17.8 (7.7, 48.7) | 0.80 |
| IL-8 (pg ml⁻¹)* | 15.4 (9.2, 27.5) | 15.1 (9.1, 26.8) | 35.7 (13.1, 129.0) | <0.0001 | 15.6 (9.3, 27.6) | 15.2 (9.1, 26.4) | 23.7 (16.4, 38.8) | 0.0035 |
| IL-10 (pg ml⁻¹)* | 19.3 (10.5, 43.6) | 19.1 (10.3, 42.6) | 34.7 (15.5, 83.3) | <0.0001 | 20.5 (10.8, 43.6) | 20.1 (10.8, 43.6) | 23.8 (12.0, 43.4) | 0.62 |
| IP-10 (pg ml⁻¹)* | 852.5 (387.5, 1,882.0) | 857.0 (394.0, 1,895.0) | 745.0 (248.0, 1,381.0) | 0.023 | 727.0 (368.0, 1,858.0) | 754.5 (383.0, 1,929.0) | 371.0 (218.0, 692.0) | 0.0001 |
| PCT (ng ml⁻¹)* | 0.3 (0.2, 0.7) | 0.3 (0.2, 0.7) | 1.0 (0.4, 6.1) | <0.0001 | 0.4 (0.2, 1.1) | 0.4 (0.2, 1.1) | 0.7 (0.4, 1.6) | 0.0072 |
| sTNF-R1 (pg ml⁻¹)* | 1,559.0 (1,244.0, 2,022.0) | 1,546.0 (1,240.0, 1,992.0) | 2,166.0 (1,549.0, 3,542.0) | <0.0001 | 1,643.5 (1,306.0, 2,153.0) | 1,627.0 (1,297.0, 2,129.0) | 1,947.0 (1,542.5, 2,905.0) | 0.0012 |
| sTREM1 (pg ml⁻¹)* | 227.0 (165.0, 333.0) | 224.0 (163.0, 322.0) | 415.0 (267.0, 615.0) | <0.0001 | 238.0 (182.0, 331.0) | 235.0 (181.0, 325.0) | 359.0 (279.5, 453.0) | <0.0001 |
| suPAR (ng ml⁻¹)* | 4.1 (3.3, 5.0) | 4.0 (3.3, 5.0) | 5.1 (3.8, 7.9) | <0.0001 | 4.6 (3.7, 5.8) | 4.5 (3.7, 5.7) | 5.6 (4.6, 7.2) | 0.0001 |
| Other laboratory biomarkers | | | | | | | | |
| Lactate (mmol l⁻¹)* | 1.0 (0.7, 1.3) | 1.0 (0.7, 1.3) | 1.1 (0.7, 1.8) | 0.0060 | 1.2 (0.9, 1.6) | 1.1 (0.8, 1.6) | 1.8 (1.3, 2.8) | <0.0001 |
| Glucose (mmol l⁻¹)* | 5.4 (4.8, 6.2) | 5.4 (4.8, 6.1) | 6.0 (5.1, 7.1) | <0.0001 | 5.5 (4.9, 6.4) | 5.5 (4.9, 6.4) | 5.9 (5.0, 7.1) | 0.14 |
| Hb (g dl⁻¹)* | 11.5 (10.6, 12.4) | 11.5 (10.6, 12.4) | 10.9 (9.5, 12.2) | 0.0019 | 10.8 (9.9, 11.6) | 10.8 (9.9, 11.6) | 10.5 (9.4, 11.5) | 0.23 |
| Recruitment strata | | | | | | | | |
| Inpatient | 2,080 (81%) | 1,983 (80%) | 97 (100%) | <0.0001 | 644 (78%) | 608 (77%) | 36 (100%) | 0.0012 |
| Outpatient | 501 (19%) | 501 (20%) | 0 | | 180 (22%) | 180 (23%) | 0 | |

CRT, capillary refill time; HAZ, height-for-age z-score; Hb, hemoglobin; LRTI, lower respiratory tract infection; SpO₂, oxygen saturation; URTI, upper respiratory tract infection; WHZ, weight-for-height z-score. Data are shown as median (Q1, Q3) or n (%) unless otherwise indicated. P values (all two-sided) were obtained by Wilcoxon rank-sum test (continuous variables), Pearson's $\chi^2$ test or Fisher's exact test (categorical variables). *Missing data: recent hospitalization, n=11 (7 derivation cohort: 6 non-severe and 1 severe; 4 validation cohort: 4 non-severe); wasted, n=12 (8 derivation cohort: 8 non-severe; 4 validation cohort: 3 non-severe and 1 severe); stunted, n=4 (2 derivation cohort: 2 non-severe; 2 validation cohort: 2 non-severe); MUAC-for-age z-score, n=216 (158 derivation cohort: 134 non-severe and 24 severe; 58 validation cohort: 41 non-severe and 17 severe); WHO danger signs, n=7 (6 derivation cohort: 5 non-severe and 1 severe; 1 validation cohort: 1 non-severe); intractable vomiting, n=4 (3 derivation cohort: 3 non-severe; 1 validation cohort: 1 non-severe); convulsions, n=5 (4 derivation cohort: 3 non-severe and 1 severe; 1 validation cohort: 1 non-severe); lethargy, n=6 (6 derivation cohort: 5 non-severe and 1 severe); heart rate, n=1 (1 derivation cohort: 1 non-severe); respiratory rate, n=2 (1 derivation cohort: 1 non-severe; 1 validation cohort: 1 non-severe); oxygen saturation, n=205 (150 derivation cohort: 110 non-severe and 40 severe; 55 validation cohort: 35 non-severe and 20 severe)**; axillary temperature, n=1 (1 derivation cohort: 1 non-severe); ANG-1, ANG-2, sFLT-1, IL-6, IL-8, IL-10, IP-10, PCT, sTNF-R1 and sTREM1, n=91 (57 derivation cohort: 51 severe and 6 non-severe; 34 validation cohort: 34 non-severe); CHI3L1, n=93 (59 derivation cohort: 52 non-severe and 7 severe; 34 validation cohort: 34 non-severe); CRP, n=96 (62 derivation cohort: 56 non-severe and 6 severe; 34 validation cohort: 34 non-severe); IL-1ra, n=92 (58 derivation cohort: 52 non-severe and 6 severe; 34 validation cohort: 34 non-severe); suPAR, n=124 (90 derivation cohort: 82 non-severe and 8 severe; 34 derivation cohort: 34 non-severe); lactate, glucose, n=98 (98 validation cohort: 91 non-severe and 7 severe); and Hb, n=574 (565 derivation cohort: 552 non-severe and 13 severe; 9 validation cohort: 9 non-severe). **For 125/205, immediate routinely collected SpO₂ on room air measurements were available and utilised for the clinical prediction models. [a]Overnight admission in the last 6 months. [b]Calculated in children aged 3–60 months (R package: zscorer)[57]. [c]Receipt of any treatment for the current illness in the community prior to presentation at the study site (of these, none had been admitted and 193 had received parenteral treatment). [d]Travel time to the study site ≤1 hour. [e]Presenting syndromes were not mutually exclusive. [f]Assessed using the Alert Voice Pain Unresponsive (AVPU) scale.

clinical model, the pulse oximetry model was better able to rule in children at risk of disease progression (positive likelihood ratio (PLR) 150.7 (95% CI: 72.9–278.2) versus 37.5 (95% CI: 12.2–121.1)), with one in three children who would have been recommended for referral developing severe disease. It also demonstrated superior rule-out performance (sensitivity 88.9% (95% CI: 76.7–97.8) and negative likelihood ratio (NLR) 0.11 (95% CI: 0.02–0.25)). Overall, the pulse oximetry model demonstrated superior rule-out properties at the rule-out threshold (difference in sensitivity = 13.7% (95% CI: 3.4–27.3)) and rule-in properties at the rule-in threshold (difference in specificity = 0.7% (95% CI 0.1–2.1)), compared to the clinical model (Extended Data Table 1).

Compared to the clinical model, the sTREM1 model improved rule-out performance to a similar extent as the pulse oximetry model (sensitivity 89.2% (95% CI: 76.9–97.5) and NLR 0.12 (95% 0.03–0.25)) but did not improve rule-in performance (PLR 18.2 (95% CI: 7.2–78.9) versus 37.5 (95% CI: 12.2–121.1)). Differences in sensitivity and specificity between the clinical model and the sTREM1 model were 13.9% (95% CI: 3.4–26.7) and −1.1% (95% CI: −2.8 to −0.03), respectively

(Extended Data Table 1). The combined model (including both SpO₂ and sTREM1) did not offer advantage over the pulse oximetry model, in terms of neither discharge (rule-out) nor referral (rule-in) performance.

All four models outperformed WHO danger signs in terms of rule-out (sensitivity 55.5% (95% CI: 39.4–72.7 and NLR 0.54 (95% CI: 0.32–0.74)) and rule-in (specificity 82.6% (95% CI: 77.1–87.6)) and PLR 3.2 (95% CI: 2.1–4.9)) performance. Differences in sensitivity and specificity between the WHO danger signs and the clinical model were 19.0% (95% CI: 0.0–41.1) and 16.4% (95% CI: 11.7–21.7), respectively (Extended Data Table 1).

**Impact analysis**

In a deterministic impact analysis applied to a hypothetical population of 10,000 children, using fixed point estimates of sensitivity, specificity and disease prevalence obtained from the validation cohort, we projected that, compared to using the clinical model, either the pulse oximetry model or the sTREM1 model would have prevented one additional child who would progress to life-threatening infection

**Table 2 | Selected variables, discrimination and calibration of the clinical prediction models in the validation cohort**

| Model | Constituent variables | Discrimination | Calibration | |
|---|---|---|---|---|
| | | AUC (95% CI) | Intercept (95% CI) | Slope (95% CI) |
| Clinical | Heart rate (bpm)<br>Respiratory rate (bpm)<br>Prostration<br>Intractable vomiting<br>Altered mental state<br>Prolonged capillary refill time | 0.95 (0.92–0.97) | 0.31 (−0.08 to 0.71) | 1.46 (1.19–1.73) |
| Pulse oximetry | Heart rate (bpm)<br>Respiratory rate (bpm)<br>Prostration<br>Intractable vomiting<br>Convulsions<br>Oxygen saturation (%) | 0.97 (0.96–0.99) | 0.60 (0.21–0.99) | 1.74 (1.41–2.06) |
| sTREM1 | Heart rate (bpm)<br>Respiratory rate (bpm)<br>Prostration<br>Intractable vomiting<br>Altered mental state<br>sTREM1 (pg ml$^{-1}$) | 0.96 (0.93–0.98) | 0.18 (−0.26 to 0.62) | 1.33 (1.04–1.63) |
| Combined | Heart rate (bpm)<br>Respiratory rate (bpm)<br>Prostration<br>Intractable vomiting<br>Oxygen saturation (%)<br>sTREM1 (pg ml$^{-1}$) | 0.98 (0.96–0.99) | 0.48 (0.05–0.91) | 1.58 (1.21–1.96) |

Variables selected in the different clinical prediction models (clinical model: clinical parameters only; pulse oximetry model: clinical parameters plus SpO$_2$; sTREM1 model: clinical parameters plus sTREM1; combined model: clinical parameters plus SpO$_2$ and sTREM1). Performance of each model in the validation cohort is summarized using its weighted discrimination and calibration. Coefficients and odds ratios for the constituent variables are reported in the appendix (Supplementary Table 6).

being discharged for every approximately 2,300 children tested (number needed to test (NNT)). For the pulse oximetry model, these gains would have been achieved in addition to a threefold reduction in the referral rate (0.3% versus 1.0%), fewer children being recommended for monitoring (5.0% versus 8.4%) and more recommended for discharge (94.7% versus 90.6%). For the sTREM1 model, the gains would have been partially offset by twice the number of referrals (2.1% versus 1.0%) compared to the clinical model.

Compared to WHO danger signs, the prediction models could prevent one additional child who would progress to life-threatening infection being discharged for every 1,000 (pulse oximetry, sTREM1 or combined models) or 1,750 (clinical model) children tested (NNT) while simultaneously substantially reducing referral rates from 17.0% to 0.3–2.1%. As WHO danger signs provide a binary classification (refer or not), any benefits gained through use of the prediction models would be partially offset by the costs of monitoring up to 8.4% of all presentations.

### Categorical outcome scale
All four models predicted increasing probability of progression to severe febrile illness at higher levels of the categorical outcome scale (Supplementary Table 7) in the validation cohort (Fig. 2), suggesting that they would be able to provide discrimination across the severity spectrum rather than only identify the most severely unwell patients.

Using the categorical outcome scale to assess triage correctness indicated that improvement in sensitivity of the other models compared to the clinical model was driven by increases in both referrals and monitoring of the most severe (category IV) patients (Fig. 3). The reduction in referrals achieved by the pulse oximetry model was a consequence of both more discharges and fewer cases being recommended for monitoring among the least unwell children (category I).

### Missed cases of severe febrile illness
Among the 133 participants who progressed to severe febrile illness, the same children would have been missed by all four models in both

the derivation and validation cohorts (Extended Data Fig. 2 and Supplementary Table 8). Furthermore, the additional participants who would have been identified by the pulse oximetry, sTREM1 and combined models were broadly consistent across these three models.

Older children, males, those with a shorter illness prior to presentation and those without pneumonia may be more likely to be missed by the models (Supplemementary Table 8). Baseline biomarker concentrations were typically less deranged in children whom the models would have failed to identify, apart from ANG-1, CRP, IL-6 and IP-10 concentrations, which were more abnormal in children who progressed to severe disease but who would have been missed by the models.

The models may be more likely to identify children who progress to severe illness quickly. Extending the definition of severe febrile illness to include all cases that developed in the 28 days after enrollment identified an additional 10 participants (two deaths and eight survivors who required organ support). Among the 42 children who progressed to develop severe febrile illness more than 24 hours after enrollment, 12 (28.6%) would have been missed by the clinical model, and the pulse oximetry and sTREM1 models would have missed nine (21.4%) and 10 (23.8%) participants, respectively.

### Cost-effectiveness
All four models were predicted to dominate (that is, were cost-saving and more effective than) WHO danger signs. Compared to the clinical model, the pulse oximetry, sTREM1 and combined models were all predicted to be cost-effective using both cost-effectiveness thresholds (CETs): $2,551 per disability-adjusted life year (DALY) averted and $459 per DALY averted (Table 3)[33–35]. Cost-effectiveness for the combined and pulse oximetry models was predicted to improve in contexts with higher referral costs, as a result of their better specificity compared to the clinical model. Although specificity for the sTREM1 model was inferior to the clinical model, it was predicted to remain cost-effective up to a referral cost of approximately $600 per patient, even when using the conservative CET (Supplementary Table 9). Neither the sTREM1 model nor the combined model was predicted to be more cost-effective than the pulse oximetry model at either of the analyzed CETs.

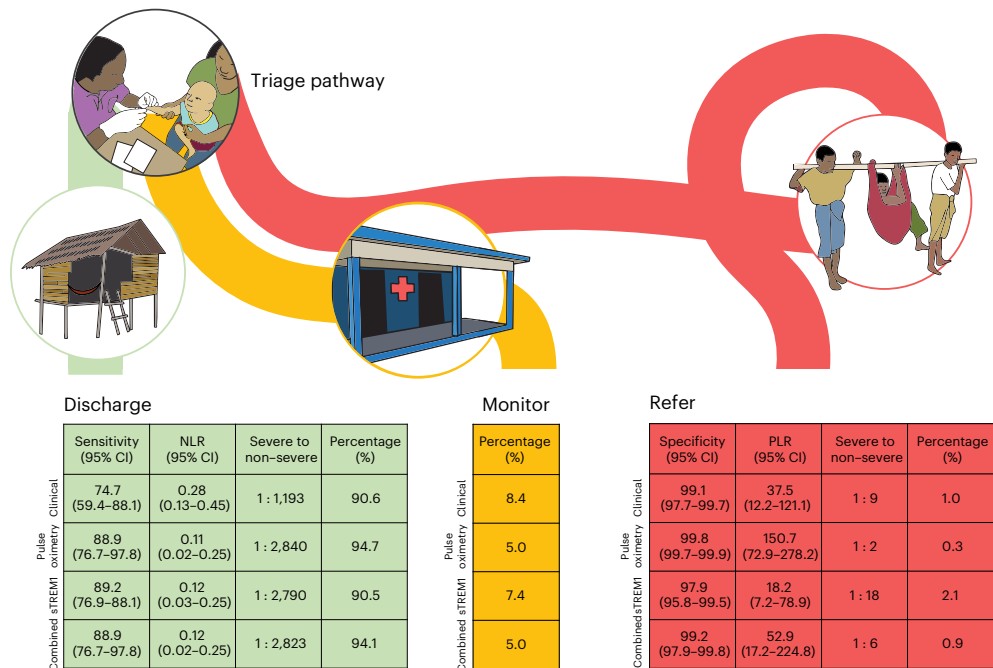

**Fig. 1 | Utility of the clinical prediction models within a hypothetical traffic light triage framework.** Among patients recommended for discharge (green; predicted probability (pp) of severe disease <0.5%), the sensitivity and NLR indicate the ability of the model to rule out severe disease. The percentage of patients recommended for monitoring (amber; pp 0.5–2%) indicates the potential burden on the health system created by the resources required for observation or proactive safety-netting. Among patients recommended for referral (red; pp >2%), the specificity and PLR indicate the ability of the model to rule in severe disease, and the percentage of patients indicates the potential burden on the health system created by the resources required for referral. Clinical model: clinical parameters only; pulse oximetry model: clinical parameters plus $SpO_2$; sTREM1 model: clinical parameters plus sTREM1; combined model: clinical parameters plus $SpO_2$ and sTREM1. NLR, negative likelihood ratio; PLR, positive likelihood ratio.

## Discussion

This multicountry prospective study, conducted in seven locations across Asia, developed and validated clinical prediction models that outperform the current standard of care for triage of febrile children in resource-constrained community settings. Although all models were superior to WHO danger signs, integration of pulse oximetry or host biomarker testing alongside measurement of simple clinical parameters was required to adequately rule out impending critical illness, a key consideration for effective risk stratification at the community level. Notably, our analyses indicate that both strategies are likely to be cost-effective across a range of CETs, including in conflict-affected settings and remote contexts, where referral costs are often high.

Our findings align with previous studies, which identified the prognostic potential of specific clinical parameters in pediatric febrile illness[36–42], and build upon emerging evidence supporting hypoxemia and host biomarkers, such as sTREM1, as indicators of poor outcome[17,20,25,43,44]. The superior performance of the prediction models compared to previous univariate analyses underscores the importance of a multivariate approach to risk stratification[30,41,45], whereby combinations of multiple 'nearly' signs can prompt action even if no 'absolute' red flags are present[46].

The pulse oximetry model outperformed the sTREM1 model and may be the preferred option in many settings. Its adoption would be particularly suited to primary health centers and rural clinics that have access to safe storage, secure supply chains, supplemental oxygen therapy and staff trained to manage other conditions for which a pulse oximeter may provide benefit. By contrast, for lesser-trained community health workers, who typically have a limited scope of practice and are accustomed to screening all febrile children for malaria, a triage tool incorporating host biomarker testing may be more practical, particularly if multiplexed with a rapid diagnostic test for malaria[31,47].

It is important to understand which patients at risk of severe illness may be missed by a particular triage approach. Our models appear to perform best in younger children with pneumonia—a group that carries a disproportionate share of the disease burden, indicating opportunity for substantial public health impact[1]. Conversely, children presenting earlier in their illness or deteriorating later may be more likely to be missed, suggesting a need for adjusted approaches to identify these children. Previous work showed that host biomarkers may be particularly valuable for identifying children whose illness severity is not clinically evident at presentation and who are at risk for delayed clinical deterioration[30,48]. In our study, both the pulse oximetry and sTREM1 models improved identification of such children relative to the clinical model, although small event numbers in this subgroup mean that further work is needed.

To minimize outcome misclassification, we applied a stringent definition of severe illness—restricted to children who died or received organ support within 2 days of enrollment—to develop the prediction models. Consequently, the reported NNTs and severe to non-severe referral ratios represent conservative estimates, emphasizing model performance in identifying children who progressed to the most severe illness. In practice, however, a substantially larger proportion of children would merit hospital referral. Secondary analyses using the categorical outcome indicate that the models provide discrimination across the severity spectrum, and further work is underway to quantify this performance.

This study developed and validated risk stratification tools for febrile children presenting directly from the community. We overcame weaknesses of previous studies by adhering to best-practice methodology for clinical prediction model building, prespecifying candidate predictors, avoiding overfitting and using a held-out geographic validation dataset to assess model performance[49]. Other strengths include eligibility criteria and a recruitment strategy that ensured a

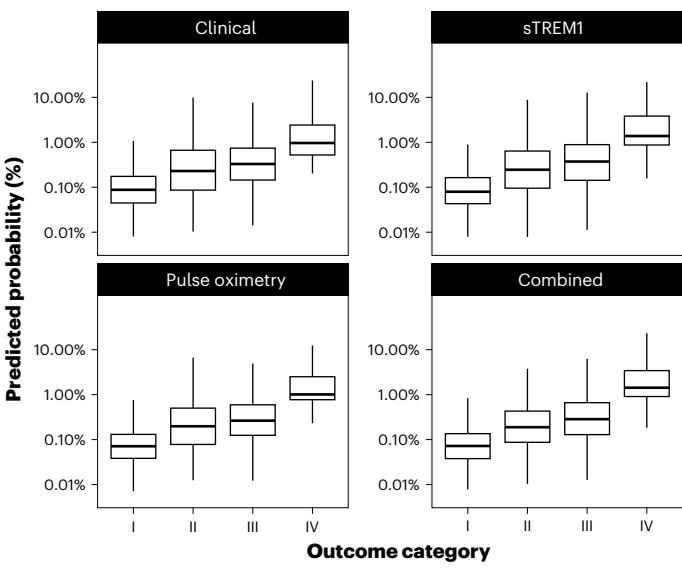

**Fig. 2 | Predicted probability of severe febrile illness across the categorical outcome scale for each prediction model in the validation cohort.** Box plots show the median (center line), IQR (box; 25th–75th percentiles) and whiskers extending to the most extreme values within 1.5× the IQR. Observations beyond this range (outliers) are not shown. Clinical model: clinical parameters only; pulse oximetry model: clinical parameters plus SpO₂; sTREM1 model: clinical parameters plus sTREM1; combined model: clinical parameters plus SpO₂ and sTREM1. Category I: managed as an outpatient and recovered by day 28 (*n* = 638). Category II: admission to any health facility for two or fewer nights between enrollment and day 28 (*n* = 779). Category III: admission to any health facility for more than two nights between enrollment and day 28 (*n* = 1,829). Category IV: death or organ support within 2 days of enrollment (*n* = 133).

study population and outcome prevalence reflective of community care settings[50,51]. Our analytical framework prioritized clinical utility, health system impact and cost-effectiveness over summary measures of model performance, such as the AUC[52].

Several limitations must be discussed. Despite efforts to approximate community settings, our study population was recruited at rural hospital outpatient departments, and differences between these patients and those seeking care in more peripheral locations may remain. The relatively short interval to developing severe disease may indicate a higher baseline severity than is typical in some community care settings. Use of trained research assistants may have inflated the apparent performance of clinical parameters; in routine care settings where interobserver variability is high, it may be that incremental benefit of pulse oximetry and host biomarker testing could be even greater[13]. Respiratory presentations were highly represented in our cohort, and most children who developed severe disease required respiratory support. This may partly explain the observed benefits of pulse oximetry but, nonetheless, reflects the prevalent spectrum of common childhood infections.

Model calibration was suboptimal, likely due to differences in the prevalence of key clinical predictors, such as intractable vomiting and convulsions, between the derivation and validation cohorts. These discrepancies may reflect the well-documented interobserver variability in assessing WHO danger signs, suggesting that excluding these variables could improve model performance[13]. Although standardized assessments might mitigate this issue, in our study, trained research assistants used precise definitions; under routine care, variability is likely to be greater. Miscalibration primarily involved underestimation of risk among the highest-risk individuals; critically, the models would still have advised referral in these cases. Nonetheless, further external validation is needed, particularly given the relatively few events in the validation dataset (*n* = 36), which contributed to imprecision in

the effect estimates. We have published our full models to encourage independent validation.

In our cohort, malaria was uncommon, consistent with the evolving epidemiology of febrile illness across Asia[53]. Circulating markers of immune and endothelial activation, such as sTREM1, appear to capture convergent pathways of host response and enable risk stratification of pediatric fever syndromes irrespective of underlying etiology, including both malarial and non-malarial infections[25,54,55]. Similarly, hypoxemia has emerged as a robust predictor of adverse outcome across a spectrum of common childhood infections[17,20]. Nonetheless, evaluation of model performance in malaria-endemic regions will be essential to establish generalizability and clinical utility.

Among the clinical biomarker models, we selected the sTREM1 model for external validation because it consistently performed best across development and sensitivity analyses, retained face validity

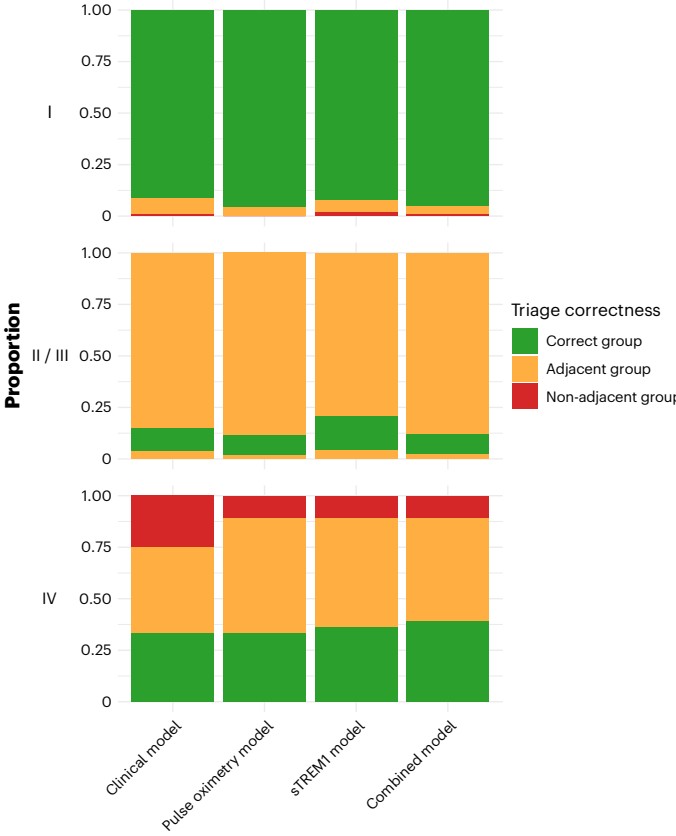

**Fig. 3 | Proportional distribution of triage groups by patient category for each prediction model.** Proportional disposition of patients in each categorical outcome category is color coded to reflect triage appropriateness: green indicates triage to correct group; amber indicates triage to group adjacent to correct group; and red indicates triage to group non-adjacent to correct group. Category I (least severe): discharged patients (predicted probability (pp) <0.5%), green; observed patients (pp 0.5–2%), amber; referred patients (pp >2%), red. Categories II and III (moderately severe): discharged patients, amber; observed patients, green; referred patients, amber. Category IV (most severe): discharged patients, red; observed patients, amber; referred patients, green. Within each outcome category (horizontal facet), patients are ordered as discharged, observed and referred, from top to bottom. Clinical model: clinical parameters only; pulse oximetry model: clinical parameters plus SpO₂; sTREM1 model: clinical parameters plus sTREM1; combined model: clinical parameters plus SpO₂ and sTREM1. Category I: managed as an outpatient and recovered by day 28. Category II: admission to any health facility for two or fewer nights between enrollment and day 28. Category III: admission to any health facility for more than two nights between enrollment and day 28. Category IV: death or organ support within 2 days of enrollment.

**Table 3 | Cost effectiveness of the pulse oximetry, sTREM1 and combined models compared to the clinical model**

| Model | Costs ($) | DALYs | Incremental costs ($) | DALYs averted | ICER ($ per DALY averted) |
|---|---|---|---|---|---|
| Clinical | 7.27 | 0.084 | - | - | - |
| Pulse oximetry | 7.95 | 0.058 | 0.69 | 0.026 | $26.28 |
| sTREM1 | 12.39 | 0.058 | 5.13 | 0.026 | $196.46 |
| Combined | 12.11 | 0.058 | 4.85 | 0.026 | $185.76 |

Clinical model: clinical parameters only; pulse oximetry model: clinical parameters plus SpO2; sTREM1 model: clinical parameters plus sTREM1; combined model: clinical parameters plus SpO2 and sTREM1.

from previous studies[25,27,30,44] and is under active development as a point-of-care assay[56]. Nonetheless, optimism-adjusted performance was similar across several candidate biomarkers, and other clinical biomarker models may prove equally or more effective. Limiting models to a single biomarker reduced overfitting and improves feasibility for application on the field, but future work should examine combinations of biomarkers, balancing potential gains in accuracy against increased cost and complexity. Including biomarkers found to be elevated in children who progressed to severe illness but who would have been missed by the current models (for example, ANG-1, CRP, IL-6 or IP-10) may be a fruitful approach.

We envisage these models being applied selectively to children for whom referral decisions are borderline and no overt signs of severe illness are evident at presentation. The exact 'red flags' triggering referral will differ across settings and are influenced by health worker capacity, safety-netting options, treatments available in the community, opportunity costs of referral and the quality of care at the higher-level facility. In our cohort, some children may be considered as presenting with features mandating referral and would not be appropriate candidates for model-based triage in all settings. Careful specification of referral criteria and further refinement and contextualization of the models will be essential prior to evaluation in clinical trials or integration into routine care.

The traffic light framework we used to illustrate clinical utility reflects the reality that, in most settings, some sort of safety-netting is possible[46]. Decision thresholds were selected pragmatically based on field experience but will require adaptation to local contexts. Future work should evaluate how different decision thresholds influence model performance and health system burden. Our results indicate that incorporating either pulse oximetry or host biomarker testing are likely to be cost-effective approaches to triage; however, the modeling outputs are inherently limited by assumptions—for example, that children recommended for monitoring and who develop severe disease can be recognized and escalated quickly—and assessing real-world impact will require interventional trials.

These findings should be interpreted in the context of the ongoing epidemiological transition in South and Southeast Asia. As malaria incidence declines, childhood febrile illness has become increasingly heterogeneous and is seldom attributable to a single, easily diagnosable and treatable cause. At first presentation, whether in primary care clinics, rural facilities, community-based services or conflict-affected settings, triage strategies must adapt to better identify children at risk of clinical deterioration. In such contexts, referral decisions entail substantial opportunity costs, yet failure to recognize severe illness early can be fatal. Strengthening prognostic capacity is, therefore, central to reinforcing primary care and advancing universal health coverage, with the potential to improve equity, reduce avoidable referrals and ensure timely escalation for the children most likely to benefit.

In summary, we developed and validated clinical prediction models that substantially improve risk stratification of febrile children compared to current practice. Incorporation of pulse oximetry or

sTREM1 measurements alongside simple clinical parameters achieves triage performance that may be acceptable to patients, providers and policymakers. Both approaches appear cost-effective across diverse contexts, including those with high referral costs. Further external validation and interventional studies will be essential to confirm clinical impact and support policy adoption.

## Online content

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

**Arjun Chandna**[1,2,3] ✉, **Constantinos Koshiaris**[4,5], **Raman Mahajan**[6], **Riris Adono Ahmad**[7], **Dinh Thi Van Anh**[8], **Khalid Shams Choudhury**[6], **Suy Keang**[1,9], **Nguyen The Nguyen Phung**[10], **Sayaphet Rattanavong**[11], **Souphaphone Vannachone**[11], **Spot Sepsis Investigator Group*,** **Chris Painter**[2,11,12], **Mikhael Yosia**[6], **Naomi Waithira**[2,12], **Mohammad Yazid Abdad**[2,12], **Janjira Thaipadungpanit**[12,13], **Paul Turner**[1,2], **Phan Huu Phuc**[8], **Dinesh Mondal**[14], **Mayfong Mayxay**[2,11,15], **Bui Thanh Liem**[10], **Elizabeth A. Ashley**[2,11], **Eggi Arguni**[7], **Rafael Perera-Salazar**[4], **Melissa Richard-Greenblatt**[2,16,17], **Yoel Lubell**[2,12,18,29] & **Sakib Burza**[3,6,19,29]

[1]Cambodia Oxford Medical Research Unit, Angkor Hospital for Children, Siem Reap, Cambodia. [2]Centre for Tropical Medicine and Global Health, University of Oxford, Oxford, UK. [3]Clinical Research Department, London School of Hygiene and Tropical Medicine, London, UK. [4]Department of Primary Care Health Sciences, University of Oxford, Oxford, UK. [5]Department of Primary Care and Population Health, University of Nicosia Medical School, Nicosia, Cyprus. [6]Médecins Sans Frontières Operational Centre Barcelona, Barcelona, Spain. [7]Centre for Tropical Medicine, Universitas Gadjah Mada, Yogyakarta, Indonesia. [8]Viet Nam National Children's Hospital, Hanoi, Vietnam. [9]Angkor Hospital for Children, Siem Reap, Cambodia. [10]University of Medicine and Pharmacy at Ho Chi Minh City, Ho Chi Minh City, Vietnam. [11]Lao-Oxford-Mahosot Hospital-Wellcome Trust Research Unit, Mahosot Hospital, Vientiane, Laos. [12]Mahidol Oxford Tropical Medicine Research Unit, Mahidol University, Bangkok, Thailand. [13]Faculty of Tropical Medicine, Mahidol University, Bangkok, Thailand. [14]International Centre for Diarrhoeal Disease Research, Dhaka, Bangladesh. [15]Institute for Research and Education Development, University of Health Sciences, Vientiane, Laos. [16]Department of Laboratory Medicine and Pathobiology, University of Toronto, Toronto, Ontario, Canada. [17]Department of Pediatric Laboratory Medicine, The Hospital for Sick Children, Toronto, Ontario, Canada. [18]Amsterdam Institute of Global Health and Development, Amsterdam, The Netherlands. [19]Health In Harmony, Portland, OR, USA. [29]These authors contributed equally: Yoel Lubell, Sakib Burza. *A list of authors and their affiliations appears at the end of the paper. ✉e-mail: arjun@tropmedres.ac

## Spot Sepsis Investigator Group

**Mohammad Yazid Abdad**[2,12], **Riris Adono Ahmad**[7], **Dinh Thi Van Anh**[8], **Eggi Arguni**[7], **Elizabeth A. Ashley**[2,11], **Elizabeth M. Batty**[2,12], **Stuart D. Blacksell**[2,12], **Latsaniphone Boutthasavong**[11], **Sakib Burza**[3,6,19,29], **Arjun Chandna**[1,2,3], **Ngoun Chanpheaktra**[9], **Khalid Shams Choudhury**[6], **Tran Quoc Dat**[8], **Vu Quoc Dat**[20], **Nicholas P. J. Day**[2,12], **Arjen M. Dondorp**[2,12], **Prakash Ghosh**[14,21,22], **Carolina Jimenez**[6], **Kevin Kain**[16], **Muhammad Karyana**[23], **Suy Keang**[1,9], **Sommay Keomany**[24], **Rungnapa Khamboocha**[12], **Constantinos Koshiaris**[4,5], **Khamfong Kunlaya**[11], **Estrella Lasry**[6], **Bui Thanh Liem**[10], **Nguyen Huy Luan**[10], **Yoel Lubell**[2,12,18,29], **Raman Mahajan**[6], **Saysamone Malavong**[25], **Mayfong Mayxay**[2,11,15], **Chonticha Menggred**[12], **Dinesh Mondal**[14], **Phung Nguyen The Nguyen**[10], **Chris Painter**[2,11,12], **Rafael Perera-Salazar**[5], **Chom Phaiphichit**[11], **Chanthala Phamisith**[25], **Phan Huu Phuc**[8], **Tiengkham Pongvongsa**[26], **Sayaphet Rattanavong**[11], **Michael Rekart**[6], **Melissa Richard-Greenblatt**[2,16,17], **Bran Sambou**[1], **Mohammad Shomik**[14], **Phouthalavanh Souvannasing**[24], **Phattaranit Tanunchai**[12], **Janjira Thaipadungpanit**[12,13], **Watcharintorn Thongpiam**[12], **Bang Huyen Tran**[27], **Claudia Turner**[1,2,9], **Paul Turner**[1,2], **Souphaphone Vannachone**[11], **Asama Vinitsorn**[12], **Ranitha Vongpromek**[12], **Manivanh Vongsouvath**[11], **Naomi Waithira**[2,12], **James A. Watson**[2,12], **Mikhael Yosia**[6] & **Asri Yuniastuti**[28]

[20]Hanoi Medical University, Hanoi, Vietnam. [21]Institute of Animal Hygiene and Veterinary Public Health, Leipzig University, Leipzig, Germany. [22]Technische Universität Berlin, Berlin, Germany. [23]INA-RESPOND, Ministry of Health Republic Indonesia, Jakarta, Indonesia. [24]Salavan Provincial Hospital, Salavan, Laos. [25]Savannakhet Provincial Hospital, Savannakhet, Laos. [26]Savannakhet Provincial Health Office, Savannakhet, Laos. [27]Oxford University Clinical Research Unit, Ho Chi Minh City, Vietnam. [28]Wates District Hospital, Yogyakarta, Indonesia.

## Methods

### Study design and setting

Spot Sepsis was a multicountry, prospective cohort study conducted in seven hospitals across Bangladesh, Cambodia, Indonesia, Laos and Vietnam[58]. Study sites were purposefully chosen to represent facilities serving rural populations and providing a first point of access to the formal healthcare sector, as a proxy for the ultimate intended use-case in community care settings. Previous analyses reported the prognostic performance of individual host biomarkers[30]. In the analyses described herein, data from six sites were used to derive the clinical prediction models, with the site in Cambodia prespecified a priori for held-out external geographic validation. This split was pragmatic, based on the ability to conduct recruitment for longer in Cambodia.

### Study site profiles

Sites located outside major urban cities were proactively approached. As part of the shortlisting process, sites were asked to indicate the proportion of their patient population residing in a rural location and the proportion of children using the hospital as a first point of contact with the formal healthcare sector. Where possible, routinely collected data were used to inform site selection. If these were not available, local clinicians and hospital administrators were asked to provide their best estimate.

Due to disruptions caused by the COVID-19 pandemic, site activation was delayed, and recruitment proceeded slower than anticipated at all sites. After 18 months of recruitment, a decision was taken to identify an additional site to boost recruitment and ensure sufficient outcome events. The second Vietnam site was subsequently identified and brought online in December 2021, acknowledging that it departed from the rural site target profile established at the outset of the study. A sensitivity analysis excluding this site was prespecified to address this.

**Bangladesh—recruitment period: 18 March 2021 to 27 April 2022.** Goyalmara Mother and Child Hospital is a non-governmental hospital managed by Médecins sans Frontières (MSF) located in Cox's Bazaar, Chattogram Division, in eastern Bangladesh, which predominantly provides health services to the forcibly displaced Rohingya refugee population. The hospital provides primary and secondary care, with approximately 20,000 outpatient attendances and 4,000 inpatient admissions annually. The 12-bed high-dependency unit provides non-invasive ventilation and inotropic therapy. There is a basic on-site diagnostic laboratory.

**Cambodia—recruitment period: 5 March 2020 to 24 February 2022.** Angkor Hospital for Children is a non-governmental pediatric hospital located in Siem Reap province, northern Cambodia. The hospital provides primary to tertiary care, with approximately 80,000 outpatient attendances and 3,000 inpatient admissions annually. The 14-bed intensive care unit provides non-invasive ventilation, mechanical ventilation, inotropic therapy and peritoneal dialysis. There is an on-site diagnostic microbiology laboratory (ISO15189 accredited since 23 November 2023).

**Indonesia—recruitment period: 22 March 2021 to 22 April 2021.** Rumah Sakit Umum Daerah Wates is a government district hospital located in Yogyakarta province, Indonesia. The hospital provides primary and secondary care, with approximately 40,000 outpatient attendances and 4,000 inpatient admissions annually. The eight-bed intensive care unit provides non-invasive ventilation, mechanical ventilation, inotropic therapy and renal replacement therapy. Basic microscopy is available on-site. Culture-based microbiology is available via nearby private laboratories.

**Laos 1—recruitment period: 10 September 2020 to 30 August 2021.** Salavan Provincial Hospital is the government provincial hospital for Salavan, a predominantly rural province in southern Laos.

The hospital provides primary to tertiary care, with approximately 65,000 outpatient attendances and 12,000 inpatient admissions annually. The three-bed pediatric intensive care unit provides non-invasive ventilation, mechanical ventilation and inotropic therapy. There is an on-site diagnostic microbiology laboratory.

**Laos 2—recruitment period: 21 January 2021 to 26 August 2021.** Savannakhet Provincial Hospital is the government provincial hospital for Savannakhet, a predominantly rural province in southern Laos. The hospital provides primary to tertiary care, with approximately 7,000 outpatient attendances and 3,000 inpatient admissions annually. The 22-bed intensive care unit provides non-invasive ventilation, mechanical ventilation and inotropic therapy. There is an on-site diagnostic microbiology laboratory.

**Vietnam 1—recruitment period: 10 May 2021 to 28 October 2022.** Dong Nai Children's Hospital is the government provincial pediatric hospital for Dong Nai province in southern Vietnam. The hospital provides primary to tertiary care, with approximately 80,000 outpatient attendances and 7,000 inpatient admissions annually. The 30-bed intensive care unit provides non-invasive ventilation, mechanical ventilation, inotropic therapy and renal replacement therapy. There is an on-site diagnostic microbiology laboratory.

**Vietnam 2—recruitment period: 8 December 2021 to 4 November 2022.** Vietnam National Children's Hospital is the government national pediatric hospital, located in Ha Noi. The hospital provides primary to quaternary care, with approximately 1,200,000 outpatient attendances and 110,000 inpatient admissions annually. The 40-bed intensive care unit provides non-invasive ventilation, mechanical ventilation, inotropic therapy and renal replacement therapy. There is an on-site diagnostic microbiology laboratory.

### Study participants

Children (aged >28 days and <60 months) presenting with a febrile illness (axillary temperature ≥37.5 °C or <35.5 °C or history of fever in the preceding 24 hours)[9,59] of ≤14 days were eligible for inclusion. Exclusion criteria were prior admission to any health facility during the current illness, receipt of >15 minutes of parenteral treatment (intravenous, intramuscular or nebulized medication, intravenous fluids or supplemental oxygen) prior to screening[60], presentation within 3 days of routine immunizations, trauma as the reason for attendance or specific known comorbidities (chronic infection (for example, viral hepatitis or tuberculosis), immunosuppression (for example, HIV or oncological conditions) or active (symptomatic or currently medicated) cardiorespiratory conditions). Participants could be enrolled only once.

### Screening and enrollment

Patients were screened during daytime working hours. Screening was stratified by admission status. Admissions could occur via the emergency or outpatient departments. Inpatients were screened consecutively upon arrival at the emergency department or inpatient ward. Children admitted from the outpatient department typically did not receive treatment before arrival on the inpatient ward. Children attending the emergency department typically received treatment promptly and were either admitted or kept for a period of observation. Thus, all patients presenting to the emergency department (both those admitted and those kept for a period of observation) were considered inpatients.

Due to high outpatient numbers, outpatient screening was randomized. Outpatients to be approached were randomly allocated by computer-generated random number tables, with the preceding week's routinely collected hospital attendance data providing the sampling frame. Screening of outpatients occurred prior to assessment by the clinical team, with three outpatients targeted for recruitment each

week at each site. On rare occasions when the health worker decided to admit a child just recruited into the outpatient strata, the participant was transferred to the inpatient strata, and the next patient specified on the random number table was approached for screening.

## Data collection

Trained study personnel measured vital signs (including $SpO_2$ via pulse oximetry) and anthropometrics, assessed clinical signs (including WHO danger signs) and collected venous blood samples and nasopharyngeal swabs at enrollment (Supplementary Table 10). Demographic information and perinatal, past medical and illness histories were collected via interview with the participant's caregiver. All data were entered onto electronic case record forms using Android tablets via Open Data Kit Collect software (version 1.25.1). Participants were provided with routine care by their treating clinician. When feasible, the study supported collection and processing of peripheral blood cultures at the discretion of the clinical team.

Participants were followed-up on days 2 and 28 after enrollment, with additional follow-up on day 1 and at discharge for inpatients. Follow-up was conducted in-person if the participant remained admitted at the study site or by telephone if they had been discharged. Study monitoring was conducted by the Clinical Trials Support Group at the Mahidol Oxford Tropical Medicine Research Unit (MORU) in Bangkok, Thailand.

## Clinical parameters and host biomarkers

Clinical parameters useful for the prognostication of pediatric febrile illness were identified through systematic review of the literature[41]. Feasibility of measurement in resource-constrained community settings was considered[61], with only the most practicable parameters retained. Prioritization and standardization followed guidance set out by the Pediatric Sepsis Predictors Standardization working group[62], operationalized through standard operating procedures (SOPs) on which all study personnel were trained prior to beginning data collection. Adherence to SOPs was assessed at site initiation visits and regular monitoring visits. Study equipment was procured centrally and distributed to sites.

Host response biomarkers were shortlisted after review of the literature and consultation with domain experts (Supplementary Table 11). In primary care, where the cause of infection is typically unknown at the time of presentation, biomarkers that are predictive across a range of pathogens are essential for risk stratification. Therefore, biomarkers implicated in final common pathways to severe febrile illness and sepsis were prioritized. Endothelial activation markers included ANG-1 (refs. 26,55,63–66), ANG-2 (refs. 25,26,48,55,63–68) and soluble FLT-1 (sFLT-1; also known as soluble VEGFR-1)[25,26,44,67,68]. Immune activation markers included CHI3L1 (refs. 25,26,68), CRP[69], IL-1ra[70–72], IL-6 (refs. 28,71,73), IL-8 (refs. 28,44,70), IL-10 (ref. 28), IP-10 (also known as CXCL10)[67], procalcitonin (PCT)[29,74], soluble TNF-R1 (sTNF-R1)[25,26,68], sTREM1 (refs. 25–27,44,67,68,75–77) and soluble uPAR (suPAR)[78–81]. Lactate, glucose and hemoglobin were also included because they are easily measured using inexpensive rapid tests, are well known to clinicians, have prognostic value[41] and are recommended in pediatric sepsis guidelines[59,82,83].

## Laboratory procedures

Venous blood samples and nasopharyngeal swabs were processed immediately. Complete blood counts were conducted on-site, and peripheral blood cultures were processed at in-country laboratories. Aliquots of whole blood, EDTA plasma, fluoride/oxalate plasma and universal transport medium were stored at −20 °C or lower. These samples were then transported at −80 °C to the MORU laboratories in Bangkok, Thailand, for further analysis and biobanking.

Biomarker concentrations in EDTA plasma were measured using the Simple Plex Ella microfluidic platform (ProteinSimple)

and the suPARnostic ELISA (ViroGates), as outlined in the appendix (Supplementary Table 12). Glucose (GLUC3; Roche Diagnostics) and lactate (LACT2; Roche Diagnostics) concentrations were quantified in fluoride/oxalate plasma.

Nucleic acid was extracted from whole blood using the MagNA Pure 24 instrument and Total NA Isolation Kit (Roche Diagnostics), following the manufacturer's instructions. Real-time polymerase chain reaction (RT–PCR) multiplex assays were used to detect viral (chikungunya, dengue, Japanese encephalitis and Zika) and bacterial (*Leptospira* spp., *Orientia tsutsugamushi* and *Rickettsia* spp.) targets in whole blood. Respiratory pathogen targets were detected directly from nasopharyngeal swabs using the FilmArray RP2 panel (BioFire Diagnostics), except for samples from Cambodia, which were processed for influenza A/B and respiratory syncytial virus (RSV) using the FTD FLU/HRSV assay (Siemens). Specimens from all sites were tested using an in-house multiplex RT–PCR assay to detect SARS-CoV-2 in nasopharyngeal swabs, based on the E and N genes, as previously described[84].

## Outcome measures

The primary outcome was development of severe febrile illness within 2 days of enrollment, defined as death or receipt of organ support (mechanical ventilation, non-invasive ventilation, inotropic therapy or renal replacement therapy).

Recognizing that illness severity is a continuum, participant outcomes were also classified on a categorical scale (Supplementary Table 7). This was used to further assess the performance of the prediction models and provide insight into the relative importance of misclassifications.

## Sample size

The methods of Riley et al.[85] were followed, and, using an anticipated outcome prevalence of 1%, conservative $R^2$ Nagelkerke of 0.15 and shrinkage factor of 0.9, it was estimated that six events per parameter (EPP) would be required to derive the prediction models. The COVID-19 pandemic, declared in the same week that recruitment began at the first site, delayed initiation of other sites and slowed enrollment. Study duration (and, thus, sample size) was determined by available resources, with recruitment continuing for as long as funding allowed. Accordingly, as there were 97 events in the derivation dataset, a maximum of 16 candidate parameters could be included for development of the prediction models while minimizing the risk of overfitting. The combined model (including clinical parameters, $SpO_2$ and one host biomarker) comprised the largest number of candidate predictors (13 parameters). Thirty-six events were accrued in the validation dataset. Although clinically and statistically significant results were obtained, effect estimates are relatively imprecise. We provide our model equations to encourage further independent external validation.

## Alteration to the sample size calculation

Sample size calculations for clinical prediction model development depend, in part, on outcome prevalence, with more events required in high-prevalence settings to avoid overfitting[86]. In the published protocol, the raw (unweighted) anticipated outcome prevalence (13%) was erroneously used in the original sample size calculation, resulting in an estimate of 14 EPP for derivation of the prediction models[58]. This mistake was recognized while the study was still recruiting. At this point, and before any data were analyzed, the sample size was recalculated using the correct (weighted) anticipated outcome prevalence of 1%. This resulted in an estimate of six EPP for derivation of the prediction models.

## Statistical methods

Categorical and continuous variables were summarized using descriptive statistics for both the derivation and validation cohorts and

compared between participants according to their outcome status, using the Wilcoxon rank-sum test, Pearson's $\chi^2$ test or Fisher's exact test as appropriate. Site-specific outpatient weights were applied to adjust for unequal probabilities of selection in the sample, arising due to random sampling of outpatients (Supplementary Table 13), to ensure that the study population was representative of all eligible children presenting to the hospital and that the outcome prevalence reflected that which might be observed in community care settings[50,51].

Candidate predictors were prespecified using domain knowledge and literature review (Supplementary Table 2). The relationship between each continuous candidate predictor and the primary outcome was explored by visually inspecting a smoothing curve to identify nonlinear patterns and by examining the overall calibration of the model. Transformations were considered only for serious violations of linearity that were substantially affecting model calibration. There were few missing data (≤3.6%) for all predictors except hemoglobin, where 16.9% (574/3,405) of participants had missing data, as complete blood counts were collected at the discretion of the treating clinical team. For all predictors, missing observations were replaced with their median value, conditional on outcome status.

To build the clinical model, all 11 prespecified candidate clinical predictors (excluding $SpO_2$) were entered into the model, and backward stepwise weighted logistic regression was used to identify the most parsimonious model. This process was repeated to build the pulse oximetry model, starting with all prespecified candidate clinical predictors, this time including $SpO_2$. To identify the most promising biomarker, clinical biomarker models were developed using backward stepwise weighted logistic regression with all prespecified candidate clinical predictors (with and without $SpO_2$) and each of the 17 biomarkers in turn. The models were internally validated using bootstrapping, with 800 samples being drawn with replacement, to adjust for optimism. A sensitivity analysis excluding the northern Vietnam site ($n = 612$), which departed from the ideal rural target site profile and where outpatient weighting was derived using different methodology (Supplementary Table 13), was performed. The clinical biomarker model with highest discrimination was carried forward for external validation. Overall, a total of four models (clinical model, pulse oximetry model, a clinical biomarker model and a combined model including the biomarker and $SpO_2$) were taken forward for validation in the validation cohort.

Models were applied to the validation dataset and the predicted probabilities estimated. Model performance was evaluated in terms of discrimination and calibration. Discrimination—the ability of a model to distinguish between individuals who did and did not experience the outcome—was assessed using the weighted AUC, with values ranging from 0.5 (no discrimination) to 1.0 (perfect discrimination). Calibration—the agreement between predicted and observed risks—was assessed using the weighted calibration intercept and slope. An ideal model has an intercept of 0 (with positive values indicating overall risk underestimation and negative values indicating overestimation) and a slope of 1 (where slopes >1 suggest underestimation in low-risk individuals and overestimation in high-risk individuals, and slopes <1 indicate the opposite pattern).

Clinical utility of each model was assessed within a traffic light triage framework by evaluating classification (sensitivity, specificity, NLR and PLR) at clinically relevant decision thresholds (predicted probabilities of severe febrile illness): any child with a predicted probability of developing severe disease <0.5% (the rule-out threshold) is discharged (green); those with a predicted probability >2% (the rule-in threshold) are referred for higher-level care (red); and those with predicted probabilities between 0.5% and 2% are monitored (amber). Differences in classification indices were estimated with 95% CI derived using bootstrap resampling procedures.

A deterministic impact analysis was applied to a hypothetical population of 10,000 children, using fixed point estimates of sensitivity, specificity and disease prevalence obtained from the validation cohort, to calculate projected triage dispositions and NNT to identify one additional child who would progress to develop severe disease.

Finally, for each model, the distribution of predicted probabilities and triage groups was examined in relation to the categorical outcome scale, to better understand the potential implications of misclassifications. Characteristics of participants who developed severe febrile illness but were missed by the models were explored in order to understand model vulnerabilities and determine what adjustments may lead to improved model performance.

Data management was performed using R version 4.3.2, and all analyses were performed using STATA version 18.

## Cost-effectiveness analyses

A cost-effectiveness analysis of the different models for the triage of febrile children was conducted, set in the context of Bangladesh, which was the study site most closely aligned with the intended use setting for the referral tool (Extended Data Fig. 3). The analyses were parameterized using primary data from the study and published estimates available in the literature (Supplementary Table 14)[87–92]. Two CETs were explored: $2,551 per DALY averted (approximating Bangladesh's 2023 gross domestic product per capita)[33,35] and a more conservative CET of $459 per DALY averted obtained from Woods et al.[34] (inflated to 2024 values). A sensitivity analysis was conducted to determine the maximum referral cost at which the different models might remain cost-effective, for both of the analyzed CETs.

## Inclusion and ethics statement

The study was designed collaboratively by investigators from all participating countries during an inaugural meeting co-convened by MSF and MORU and held in Bangkok, Thailand, in May 2019. During the meeting, the study hypotheses and design were discussed to ensure relevance of the research to the study settings. The draft protocol was then iteratively improved based on feedback from stakeholders, healthcare providers and the Young Persons Advisory Group at Angkor Hospital for Children, Cambodia. Throughout the study, capacity building activities were a priority and included training of research teams, tailored support for blood culture infrastructure and opportunities for junior team members to present findings at internal and external meetings. Regional capacity for biomarker assays was developed at the MORU laboratories in Bangkok, Thailand. Study progress was reviewed at monthly co-investigator meetings, where feedback from the site teams was discussed. The diversity of the study team is reflected by the authorship.

## Ethical approvals

Caregivers of all participants provided informed written consent. The study was prospectively registered on ClinicalTrials.gov (NCT04285021) and received ethical approval from the sponsors (MSF Ethical Review Board, MSF ERB 1967, and Oxford Tropical Medicine Research Committee, OxTREC 59-19) and ethical review boards in all participating countries (International Centre for Diarrhoeal Disease Research, Bangladesh, PR-200006; Angkor Hospital for Children Research Committee, Cambodia, 01296/AHC; National Ethics Committee for Health Research, Cambodia, 264/NECHR; Medical and Health Research Ethics Committee, Indonesia, KE/FK/1397/EC/2019; National Ethics Committee for Health Research, Laos, 051/NECHR; University of Medicine and Pharmacy at Ho Chi Minh City, Vietnam, 818/HDDD-DHYD; and Ethics Committee for Biomedical Research, Vietnam, VNCH-RICH-2021-77).

## Reporting summary

Further information on research design is available in the Nature Portfolio Reporting Summary linked to this article.

## Data availability

Deidentified, individual participant data from this study will be available to researchers whose proposed purpose of use is aligned with the study objectives and approved by the data access committees at MSF and MORU. Inquiries or requests for data can be sent to data.sharing@london.msf.org and datasharing@tropmedres.ac, with an anticipated timeline of 1 month between submission of a request and committee decision. Researchers interested in accessing biobanked samples should contact the corresponding author, who will coordinate with the Spot Sepsis Sample Use Committee, with an anticipated timeline of 1 month between submission of a request and committee decision.

## Code availability

Researchers interested in accessing the code should contact the corresponding author, A.C. (arjun@tropmedres.ac), who will coordinate with the study statisticians, with an anticipated timeline of 1 month between submission of a request and response.

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

## Acknowledgements

We are extremely grateful to all the participants and their caregivers and to the laboratory and research staff at the participating sites and MORU laboratories in Bangkok, Thailand, for specimen processing, data management and study monitoring. We acknowledge the support of bioMérieux, which provided laboratory consumables and equipment for some of the respiratory specimen assays. We also acknowledge the valuable support of William Robertson (MSF), without whom the development of the study would not have been possible, and guidance of N. J. White (MORU) at key points during the study and R. Western (Health In Harmony) for assistance with the production of Fig. 1. The study was co-funded by MSF Spain (S.B.) and Wellcome (219644/Z/19/Z; Y.L.). MSF maintained a sponsor/ investigator role for the study. Wellcome had no role in study design, data collection, data analysis, data interpretation, writing of the report or decision to submit for publication.

## Author contributions

A.C., C.K., R.A.A., P.T., M.M., E.A.A., E.A., R.P.-S., M.R.-G., Y.L. and S.B. conceptualized the study. A.C., D.T.V.A., K.S.C., S.K., P.N.T.N., S.R., S.V., P.H.P., D.M., B.T.L. and E.A. acquired the data. A.C., R.M., M.Y., N.W. and M.R.-G. curated the data. M.Y.A., J.T. and M.R.-G. performed the laboratory assays. A.C., C.K., R.M. and R.P.-S. did the formal analysis. Y.L. and S.B. acquired funding. A.C. wrote the original draft of the manuscript. All authors reviewed and edited the manuscript. A.C., C.K., R.M., M.R.-G., Y.L. and S.B. verified the underlying data. A.C., C.K., Y.L. and S.B. had full access to all of the data in the study and had final responsibility for the decision to submit for publication.

## Competing interests

The authors declare no competing interests.

## Additional information

**Extended data** is available for this paper at https://doi.org/10.1038/s41591-026-04338-1.

**Correspondence and requests for materials** should be addressed to Arjun Chandna.

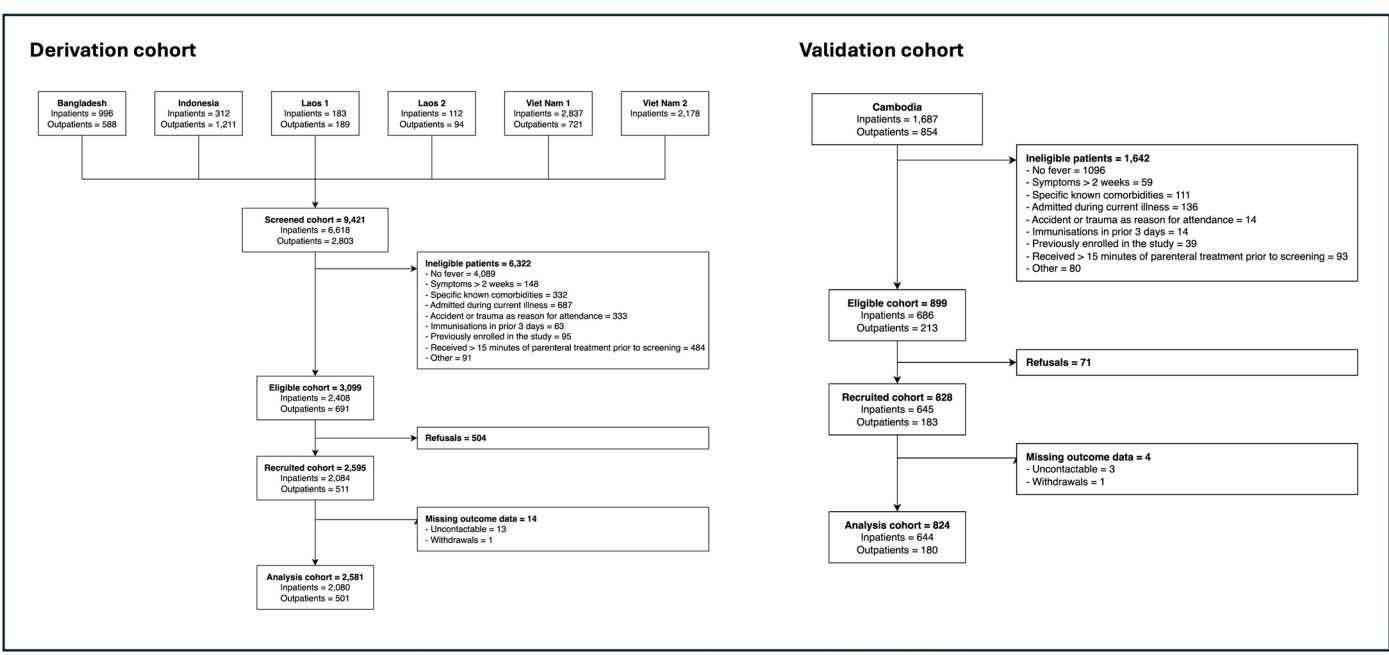

**Extended Data Fig. 1 | Study flowchart.** One reason for ineligibility is provided per patient, according to the hierarchy listed in the figure. 1,245 ineligible children had more than one reason for ineligibility (1,245/7,964; 15.6%).

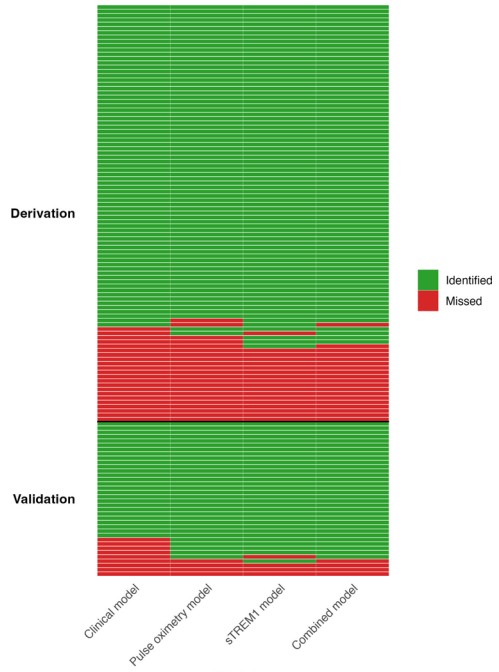

**Extended Data Fig. 2 | Identification of participants who progressed to severe febrile illness for each of the models, split by derivation and validation cohort.** Classification for each of the 133 participants who progressed to develop severe febrile illness according to whether they were identified (green; pp > 0.5%) or missed (red; pp ≤ 0.5%) by each of the models. Clinical model = clinical parameters only; pulse oximetry model = clinical parameters plus SpO2; sTREM1 model = clinical parameters plus sTREM1; combined model = clinical parameters plus SpO2 and sTREM1. pp = predicted probability of severe febrile illness.

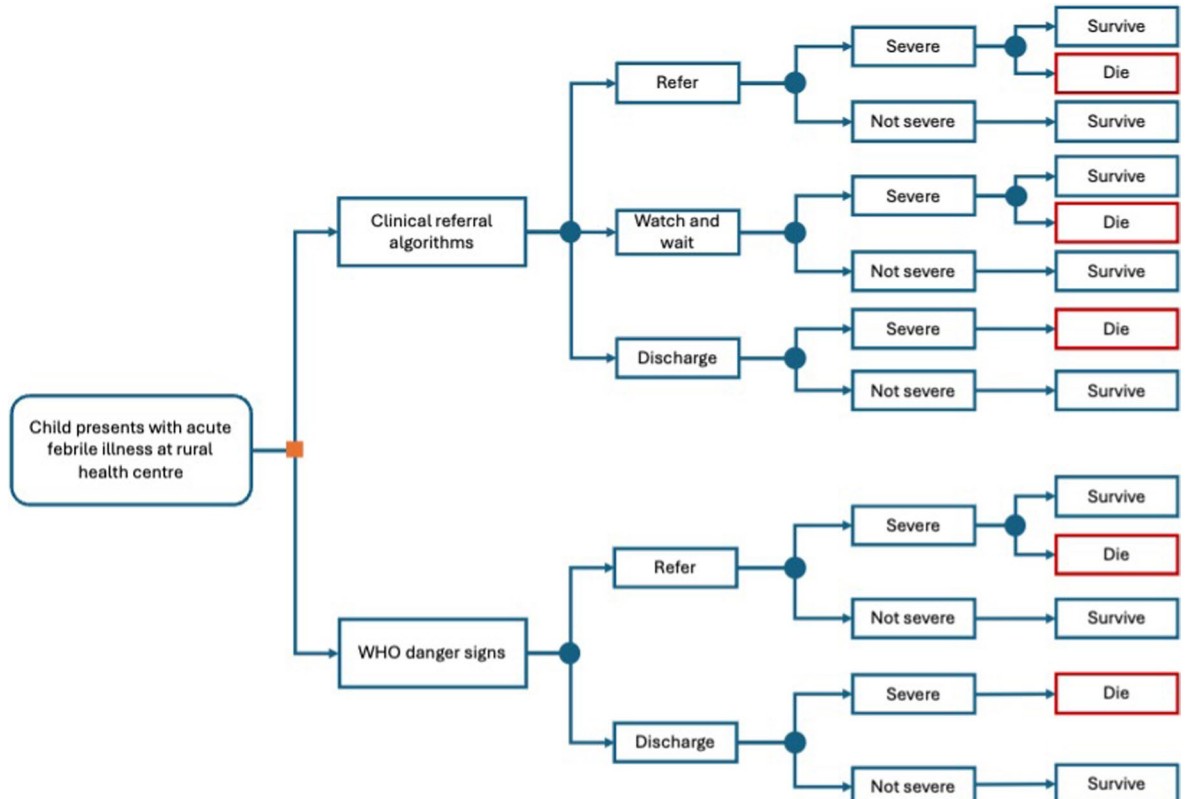

**Extended Data Fig. 3 | Cost-effectiveness analysis decision tree model structure.** Analytical framework to evaluate the comparative cost-effectiveness of the WHO danger signs vs. the clinical referral algorithms for triage of febrile children presenting from the community. Children triaged by the WHO danger signs can either be referred or discharged. Children triaged by the referral algorithms can either be referred, monitored for a period, or discharged. All children who develop non-severe febrile illness are assumed to survive. All discharged children who develop severe febrile illness are assumed to die, whereas a proportion of referred or monitored children who develop severe febrile illness are assumed to survive.

**Extended Data Table 1 | Comparison of the different triage approaches**

| | | Clinical model (95% CI) | Pulse oximetry model (95% CI) | Difference or Ratio (95% CI) |
|---|---|---|---|---|
| **Rule out** | **Sensitivity** | 74.7 (59.4 to 88.1) | 88.9 (76.7 to 97.8) | 13.7 (3.4 to 27.3) |
| | **NLR** | 0.28 (0.13 to 0.45) | 0.11 (0.02 to 0.25) | 0.43 (0.16 to 0.78) |
| **Rule in** | **Specificity** | 99.1 (97.7 to 99.7) | 99.8 (99.7 to 99.9) | 0.7 (0.11 to 2.13) |
| | **PLR** | 37.5 (12.2 to 121.1) | 150.7 (72.9 to 278.2) | 3.9 (1.3 to 12.7) |

| | | Clinical model (95% CI) | sTREM1 model (95% CI) | Difference or Ratio (95% CI) |
|---|---|---|---|---|
| **Rule out** | **Sensitivity** | 74.7 (59.4 to 88.1) | 89.2 (76.9 to 97.5) | 13.9 (3.4 to 26.7) |
| | **NLR** | 0.28 (0.13 to 0.45) | 0.12 (0.03 to 0.25) | 0.44 (0.15 to 0.81) |
| **Rule in** | **Specificity** | 99.1 (97.7 to 99.7) | 97.9 (97.7 to 99.7) | -1.1 (-2.8 to -0.03) |
| | **PLR** | 37.5 (12.2 to 121.1) | 18.2 (7.2 to 78.9) | 0.50 (0.16 to 1.19) |

| | | WHO danger signs (95% CI) | Clinical model (95% CI) | Difference or Ratio (95% CI) |
|---|---|---|---|---|
| **Rule out** | **Sensitivity** | 55.5 (39.4 to 72.7) | 74.7 (59.4 to 88.1) | 19.0 (0 to 41.1) |
| | **NLR** | 0.54 (0.32 to 0.74) | 0.28 (0.13 to 0.45) | 0.51 (0.22 to 0.93) |
| **Rule in** | **Specificity** | 82.6 (77.1 to 87.6) | 99.1 (97.7 to 99.7) | 16.4 (11.7 to 21.7) |
| | **PLR** | 3.2 (2.1 to 4.9) | 37.5 (12.2 to 121.1) | 12.5 (3.85 to 50.0) |

For sensitivity and specificity, the absolute difference between the two models with 95% CI for the difference is reported. A CI that does not cross 0, indicates a statistically significant result. For the NLR and PLR, the ratio between the two models with 95% CI for the ratio is reported. A CI that does not cross 1, indicates a statistically significant result. CI = confidence interval; NLR = negative likelihood ratio; PLR = positive likelihood ratio.

# Reporting Summary

## Statistics

For all statistical analyses, confirm that the following items are present in the figure legend, table legend, main text, or Methods section.

| n/a | Confirmed | |
|---|---|---|
| ☒ | ☐ | The exact sample size (*n*) for each experimental group/condition, given as a discrete number and unit of measurement |
| ☒ | ☐ | A statement on whether measurements were taken from distinct samples or whether the same sample was measured repeatedly |
| ☒ | ☐ | The statistical test(s) used AND whether they are one- or two-sided<br>*Only common tests should be described solely by name; describe more complex techniques in the Methods section.* |
| ☒ | ☐ | A description of all covariates tested |
| ☒ | ☐ | A description of any assumptions or corrections, such as tests of normality and adjustment for multiple comparisons |
| ☒ | ☐ | A full description of the statistical parameters including central tendency (e.g. means) or other basic estimates (e.g. regression coefficient) AND variation (e.g. standard deviation) or associated estimates of uncertainty (e.g. confidence intervals) |
| ☒ | ☐ | For null hypothesis testing, the test statistic (e.g. *F*, *t*, *r*) with confidence intervals, effect sizes, degrees of freedom and *P* value noted<br>*Give P values as exact values whenever suitable.* |
| ☒ | ☐ | For Bayesian analysis, information on the choice of priors and Markov chain Monte Carlo settings |
| ☒ | ☐ | For hierarchical and complex designs, identification of the appropriate level for tests and full reporting of outcomes |
| ☒ | ☐ | Estimates of effect sizes (e.g. Cohen's *d*, Pearson's *r*), indicating how they were calculated |

*Our web collection on statistics for biologists contains articles on many of the points above.*

## Software and code

Policy information about availability of computer code

| | |
|---|---|
| Data collection | Open Data Kit Collect version 1.25.1. |
| Data analysis | R version 4.3.2 and STATA version 18. |

For manuscripts utilizing custom algorithms or software that are central to the research but not yet described in published literature, software must be made available to editors and reviewers. We strongly encourage code deposition in a community repository (e.g. GitHub). See the Nature Portfolio guidelines for submitting code & software for further information.

## Data

Policy information about availability of data

All manuscripts must include a data availability statement. This statement should provide the following information, where applicable:
- Accession codes, unique identifiers, or web links for publicly available datasets
- A description of any restrictions on data availability
- For clinical datasets or third party data, please ensure that the statement adheres to our policy

De-identified, individual participant data from this study will be available to researchers whose proposed purpose of use is aligned with the study objectives and approved by the data access committees at MSF and MORU. Enquiries or requests for data can be sent to data.sharing@london.msf.org and datasharing@tropmedres.ac, with an anticipated timeline of one month between submission of a request and committee decision. Researchers interested in

# Research involving human participants, their data, or biological material

Policy information about studies with human participants or human data. See also policy information about sex, gender (identity/presentation), and sexual orientation and race, ethnicity and racism.

| Reporting on sex and gender | Proportion of male and female participants recruited reported in Table 1 |
|---|---|
| Reporting on race, ethnicity, or other socially relevant groupings | No reporting on race or ethnic groups performed as these vary considerably across and within different study locations |
| Population characteristics | 3,423 children aged 1-59 months presenting with acute febrile illnesses. Screening occurred during daytime working hours. Inpatients were recruited consecutively. A random selection of outpatients were recruited, informed by computer-generated random number tables which utilised the previous week's routinely collected hospital data for the sampling frame. The main biases that may have occurred as a result of the recruitment process is that the study sample may not be representative of patients presenting out of hours and that the outpatient sample may be more representative of patients attending the hospital earlier in the working day. |
| Recruitment | 5 March 2020 to 4 November 2022 |
| Ethics oversight | The study received ethical approval from the sponsors (Médecins Sans Frontières Ethical Review Board MSF ERB 1967; Oxford Tropical Medicine Research Committee OxTREC 59-19) and ethical review boards in all participating countries (International Centre for Diarrhoeal Disease Research, Bangladesh PR-200006; Angkor Hospital for Children Research Committee, Cambodia 01296/AHC; National Ethics Committee for Health Research, Cambodia 264/NECHR; Medical and Health Research Ethics Committee, Indonesia KE/FK/1397/EC/2019; National Ethics Committee for Health Research, Laos 051/NECHR; University of Medicine and Pharmacy at Ho Chi Minh City, Viet Nam; 818/HDDD-DHYD; Ethics Committee for Biomedical Research, Viet Nam VNCH-RICH-2021-77). |

Note that full information on the approval of the study protocol must also be provided in the manuscript.

# Field-specific reporting

Please select the one below that is the best fit for your research. If you are not sure, read the appropriate sections before making your selection.

☒ Life sciences          ☐ Behavioural & social sciences          ☐ Ecological, evolutionary & environmental sciences

For a reference copy of the document with all sections, see nature.com/documents/nr-reporting-summary-flat.pdf

# Life sciences study design

All studies must disclose on these points even when the disclosure is negative.

| Sample size | The methods of Riley et al. were followed and using an anticipated outcome prevalence of 1%, conservative R2 Nagelkerke of 0.15, and shrinkage factor of 0.9, it was estimated that six events per parameter would be required to derive the prediction models. The Covid-19 pandemic, declared in the same week that recruitment began at the first site, delayed initiation of other sites and slowed enrolment. Study duration (and thus sample size) was determined by available resources, with recruitment continuing for as long as funding allowed. Accordingly, as there were 97 events in the derivation dataset, a maximum of 16 candidate parameters could be included for development of the prediction models, whilst minimising the risk of overfitting. |
|---|---|
| Data exclusions | 18 participants. Lost to follow up and so no data available available on outcome status. |
| Replication | Cross-validation during the development of the prediction models. External validation of the prediction models in a held-out dataset. |
| Randomization | Not applicable - observational study. |
| Blinding | Not applicable - observational study therefore no group allocation. |

# Reporting for specific materials, systems and methods

We require information from authors about some types of materials, experimental systems and methods used in many studies. Here, indicate whether each material, system or method listed is relevant to your study. If you are not sure if a list item applies to your research, read the appropriate section before selecting a response.

## Materials & experimental systems

| n/a | Involved in the study |
|---|---|
| ☒ ☐ | Antibodies |
| ☒ ☐ | Eukaryotic cell lines |
| ☒ ☐ | Palaeontology and archaeology |
| ☒ ☐ | Animals and other organisms |
| ☐ ☒ | Clinical data |
| ☒ ☐ | Dual use research of concern |
| ☒ ☐ | Plants |

## Methods

| n/a | Involved in the study |
|---|---|
| ☒ ☐ | ChIP-seq |
| ☒ ☐ | Flow cytometry |
| ☒ ☐ | MRI-based neuroimaging |

# Clinical data

Policy information about clinical studies

All manuscripts should comply with the ICMJE guidelines for publication of clinical research and a completed CONSORT checklist must be included with all submissions.

| | |
|---|---|
| Clinical trial registration | ClinicalTrials.gov NCT04285021 |
| Study protocol | https://osf.io/v594s/ and https://bmjopen.bmj.com/content/11/1/e045826 |
| Data collection | Goyalmara Mother and Child Hospital, Bangladesh: 18/03/2021 to 27/04/2022<br>Angkor Hospital for Children, Cambodia: 05/03/2020 to 24/02/2022<br>Rumah Sakit Umum Daerah Wates, Indonesia: 22/03/2021 to 22/04/2021<br>Salavan Provincial Hospital, Lao PDR: 10/09/2020 to 30/08/2021<br>Savannakhet Provincial Hospital, Lao PDR: 21/01/2021 to 26/08/2021<br>Dong Nai Children's Hospital, Viet Nam: 10/05/2021 to 28/10/2022<br>Viet Nam National Children's Hospital, Viet Nam: 08/12/2021 to 04/11/2022 |
| Outcomes | Primary outcome: death or receipt of organ support (mechanical ventilation, non-invasive ventilation, inotropic therapy, or renal replacement therapy) within two days of enrolment. Assessed by research staff via observation at the bedside and/or interview with caregiver.<br><br>Secondary outcome: Category IV = Death or organ support ≤ 2 days after enrolment; Category III = Admission to any health facility for > 2 nights between enrolment and D28 OR death or organ support between D2 and D28; Category II = Admission to any health facility for ≤ 2 nights between enrolment and D28 OR symptoms not resolved by D28; Category I = Not admitted to any health facility between enrolment and D28 AND recovered by D28. Assessed by research staff via observation at the bedside and/or interview with caregiver. |

# Plants

| | |
|---|---|
| Seed stocks | NA |
| Novel plant genotypes | NA |
| Authentication | NA |

