## [Peer Review File · Nature Medicine]

Referral of febrile children in resource-constrained community settings in South and Southeast Asia

Corresponding Author: Dr Arjun Chandna

Version 0:

Reviewer comments:

Reviewer #1

(Remarks to the Author)

I have read your manuscript with interest. I have several suggestions to improve the quality of your manuscript.

1. Abstract: The quality of the abstract can be improved. The sensitivity and specificity of WHO Danger Signs can be de-emphasized or provided as comparator. On first glance, this reviewer mistook the reported metrics of the risk-stratification models developed and validated. Explicitly listing the cost-effectiveness metrics compared to WHO Dangers Signs is recommended.
2. Please clarify that the dataset used is the same one as the one published by your group in Lancet Child and Adolescent Health. The focus of your current manuscript being risk-prediction models for referral as opposed to a primary focus on identifying biomarker models for disease progression.
3. Figure 1 text within this figure is ineligible to this reviewer. Please increase figure dpi.
4. Clinical utility (Results): What is the confidence intervals for the estimated percentages of discharge, monitor, and referral according to each model. It is unclear to this reviewer if these differences are numerically different across models (as presented) or robust statistical measures have been used to distinguish performance of models.
5. Clinical utility estimates inform number needed to treat analyses. Again, whether these are statistically different across models is not clear to this reviewer.
6. Cost-effectiveness models can be better described in the manuscript and supplement. How were cost of monitoring patients accounted for in estimates?
7. The discussion section can be made more succinct.
8. In several instances, results are presented as though they are reporting results from prospective testing of models. The manuscript would benefit from adjusting language to indicate that what is being presented is predicted effect of risk-models if implemented in the real-world.

(Remarks on code availability)

Reviewer #2

(Remarks to the Author)

Comments to authors: The authors of the study developed and validated a clinical prediction model to guide referral of febrile children in resource limited settings. I would like to congratulate the authors for this extraordinary study which could significantly impact the health outcomes of children with febrile illness in low-income countries. Overall, the manuscript is

well written and methodologically sound. My main criticism is centered around the deviations from the original protocol and lack of clarity of selection of the validation cohort. My specific comments are listed below:

Methods:

- Need to cite the reference of the published study protocol (<https://pubmed.ncbi.nlm.nih.gov/33495264/>)
- Was Cambodia selected as validation cohort a priori? If that is the case the published protocol does not indicate that. The protocol indicates “a validation cohort will be geographically distinct”. It is unclear what geographically distinct cohort means.
- The original sample size for derivative and validation cohort indicated 280 and 100 outcomes, respectively. Authors need to explain how their final sample size could affect their accuracy.

Results:

- Lines 123-126: Validation cohort had higher % of chikungunya. Could this affect the prediction model?
- I am a bit surprised to see lower percentage of no focus febrile illness in these countries which are frequent due to dengue/chikungunya/scrub typhus etc. Is there possibility of selection bias in their enrolment process?
- Could authors add organ support breakdown (number of children receiving ventilation, renal replacement etc.). This is of interest if this is mainly driven by respiratory syndrome considering the % of severe cases with LRTI/URTI and better performance of pulse oximetry model.

Discussion:

- In limitations authors should discuss the deviations from protocol and how that impact their results.

Table 1:

- Please add the admission type distribution of “outpatient” and “inpatient” status at the time of enrolment
- The presenting syndrome percentage adds to >100. Need to indicate if they are not mutually exclusive

Figure 3:

- For category ii/iii, should the referred patients be “RED”? Amber color is both for discharged and referred group. Need clarification.

(Remarks on code availability)

Reviewer #3

(Remarks to the Author)

A: Summary of the key results

The study titled “Referral of febrile children in resource-constrained community settings in Asia (Spot Sepsis) – a multi-country, prospective cohort study” by Chandna et al. investigates alternative approaches to improve the identification of febrile children < 5 years of age who are at risk of developing severe disease in a multi-centre, multi-country study design. The modelling approach is based on an impressive dataset and outputs suggested that adding pulse oximetry and/or sTREM 1 to management guidelines could outperform current WHO danger-sign criteria in predicting severe disease.

B: Originality and significance: if not novel, please include reference

The modelling strategy is innovative and represents an important step toward evidence-based refinement of referral guidelines; however, the population-level impact appears modest and the authors’ interpretation of the clinical significance may be partially overstated given the small absolute effect and the need for validation in clinical trials.

C: Data & methodology: validity of approach, quality of data, quality of presentation

Overall, the study shows exceptional quality and provides a constructive discussion of its limitations that may inspire further research in this important field. The used methodology seems appropriate and thorough. However, I suggest seeking further expertise from additional reviewers regarding the development of the model as my experience is limited in this regard.

D: Appropriate use of statistics and treatment of uncertainties

Statistics provided are appropriate, transparent and thoroughly reported.

E: Conclusions: robustness, validity, reliability

As noted above, the overall impact of the study appears overemphasised given the comparatively small absolute effect at the population level. This limitation could be more clearly acknowledged throughout the manuscript. Nevertheless, the findings are robust, methodologically sound, and have the potential to inform global guideline development, warranting further evaluation in clinical trials.

F: Suggested improvements: experiments, data for possible revision

Line 194-210: This section is critical and summarises the main study findings well while acknowledging limitations of number of outcome events. However, the article generally appears to overstate the population-level impact of the models to my understanding. Although every child’s life is invaluable, the authors should emphasise that the absolute effect remains small, with high NNT, even if the models offer improved discrimination and could serve as an adjunct to existing WHO guidelines following clinical evaluation in clinical trials.

Line 56-58: As an economic evaluation is added in this study, authors could consider adding the financial barrier as additional reason of delayed care.

Line 82: Authors should consider to omit this sentence to avoid confusion and discuss this aspect in the discussion section in respect to the comments below.

Line 88: The authors should consider to explicitly state that the prognostic models were developed using the same patient cohort in which the biomarkers' performance was previously evaluated, to ensure transparency and prevent overinterpretation. This could be added to the methods section.

Line 177: Add n/N (25%)

Line 277-281: Authors should review this suggestion as availability of oxygen supply may be limited in primary healthcare settings and rural clinics in the respective areas. This shouldn't downplay the importance of pulse oximetry; however, it should be taken in account that experience handling hypoxaemia may and probably will further be limited.

Line 281-284: Authors should review this recommendation for two reasons. Overinterpretation of malaria RDTs is a growing concern globally, particularly in sub-Saharan Africa, where over-prescription of antimalarials is driving resistance, and leads to underestimation of other diseases with potentially underlying chronic malaria infection. As no clear cut-off for the proposed biomarker is available and its clinical relevance is yet to be assessed, authors should consider softening the wording of their recommendation; otherwise introducing the proposed assays could lead to overinterpretation of "positive" tests. However, while a positive malaria RDT suggests direct action (-> antimalarials), incorporating host biomarkers into a (combined) RDT for community health workers is simplifying complex clinical decisions which is out of scope for this profession (Is initiating antimalarial treatment sufficient to prevent progression to severe disease? Does a positive sTREM-1 always indicate hospital admission? If yes, which level of care should be considered as severe disease may be more likely?).

Authors could consider discussing the performance of sTREM 1 regarding their study cohort given the distribution of pathogens. The study team identified very few bacterial infections where sTREM 1 was found to be particularly valuable in identifying severe disease to my knowledge.

G: References: appropriate credit to previous work?

The choice and use of references are thorough and appropriate for the purpose of the manuscript.

H: Clarity and context: lucidity of abstract/summary, appropriateness of abstract, introduction and conclusions

In respect to the above-mentioned aspects, abstract, introduction, and conclusions are coherent and appropriately aligned with the study objectives. The provided background is adequate enabling readers to understand the rationale and significance of the work.

(Remarks on code availability)

na

Version 1:

Reviewer comments:

Reviewer #1

(Remarks to the Author)

Thank you for addressing concerns and suggestions. I congratulate you on your contribution to the field.

(Remarks on code availability)

N/A

Reviewer #2

(Remarks to the Author)

I reviewed authors' responses to my queries. I appreciate the clarification regarding sample size adjustment and authors have appropriately acknowledged in the limitations of the study regarding the small number of events in validation cohort. Responses to my queries have been addressed. I have no further queries.

(Remarks on code availability)

Reviewer #3

(Remarks to the Author)

All reviewer comments have been adequately addressed. I congratulate the authors on their impressive work and support publication of the manuscript in its current form. I have only one minor suggestion regarding wording in the abstract.

The term "simple clinical parameters" is potentially misleading and should be reconsidered. I suggest omitting the sentence and revise the following one to:

"Including either pulse oximetry or the host biomarker sTREM-1 substantially improved identification of children at risk of severe disease (death or organ support within two days) and outperformed WHO danger signs while increasing sensitivity [...]"

(Remarks on code availability)

Editorial comments

We expect that you fully clarify the extent of conceptual advance, and sufficiently distinguish this study from the previous Lancet publication, which we expect further analyses to include new data to make this sufficiently different from the Lancet child and adolescent health study.

We are grateful for the opportunity to clarify the distinction between this manuscript and our prior publication in *The Lancet Child & Adolescent Health*.¹ The two papers address related but fundamentally different scientific questions. Our previous work focused on evaluating the predictive performance of individual host biomarkers for progression to severe febrile illness. In contrast, the present manuscript evaluates implementable clinical referral strategies – integrating biomarker testing with clinical assessment – and quantifies their clinical utility, projected health system impact, and cost-effectiveness in resource-limited primary care settings.

Existing literature, including our prior publication, has largely evaluated host biomarkers in isolation, typically using discrimination metrics such as the area under the receiver operating characteristic curve.¹⁻⁴ While informative for biomarker development, such analyses do not address whether or how these biomarkers should be deployed in practice, nor their implications for referral decisions, health system capacity, or costs.⁵ The present study therefore moves beyond diagnostic accuracy to a decision-analytic framework, using gold-standard methods to derive and externally validate multivariable clinical prediction models and to evaluate their added value compared with the current standard of care (WHO danger signs) and triage strategies that rely on simple clinical assessment with and without pulse oximetry.⁶

Importantly, this paper shifts the unit of analysis from individual predictors to implementable referral strategies. Using a field-centric, decision-analytic approach, we quantified the gains that could be realised using different triage approaches (changes in sensitivity and specificity of referral processes) and contextualised these using a deterministic impact assessment to estimate population-level consequences such as referral volumes and numbers needed to test. We further extended these analyses by incorporating cost-effectiveness modelling to explicitly account for the opportunity costs of introducing tools such as pulse oximetry and host biomarker testing in settings where referral is expensive and health system resources are constrained.

While both papers draw on the same underlying dataset, this manuscript introduces new analytical frameworks and inferential targets – including clinical utility, health system impact, and economic value – that were not examined in the prior publication and cannot be addressed through univariate biomarker analyses alone. In response to reviewer feedback, we have further strengthened this distinction by adding formal statistical comparisons of clinical utility between triage strategies, restructuring the manuscript to more clearly separate clinical utility and health system impact results, and distinguishing the scope of the two papers more clearly in the introduction.

Collectively, these elements represent a substantive conceptual advance from biomarker evaluation to decision science and implementation-relevant modelling, addressing how diagnostic innovations can be translated into improved referral processes and more efficient use of scarce health system resources.

Reviewer #1

Abstract: The quality of the abstract can be improved. The sensitivity and specificity of WHO Danger Signs can be de-emphasized or provided as comparator. On first glance, this reviewer mistook the reported metrics of the risk-stratification models developed and validated. Explicitly listing the cost-effectiveness metrics compared to WHO Dangers Signs is recommended.

Thank you for highlighting that the Abstract was not clear. We have rewritten this taking into account your suggestions. We note that including the cost effectiveness metrics has taken the word count slightly over that recommended by Nature Medicine. We hope this is acceptable to the Editors.

Please clarify that the dataset used is the same one as the one published by your group in Lancet Child and Adolescent Health. The focus of your current manuscript being risk-prediction models for referral as opposed to a primary focus on identifying biomarker models for disease progression.

Thank you for this suggestion. We have rewritten the final paragraph of the introduction to clarify the focus on the manuscript and that the same dataset has been used. In addition, taking into consideration the editorial comments, we have also used this as an opportunity to distinguish these analyses from our previous work.

“Building on our previous work and using the same multi-country dataset, our objective in this manuscript was to address a different, complementary question: how prognostic information can

be integrated into risk prediction models to guide referral decisions for febrile children in resource-constrained community contexts and decentralised models of care. Whereas our previous analysis focused on the predictive performance of individual host biomarkers for disease progression, the present study develops and validates multivariable clinical prediction models comprising simple clinical parameters (for example, vital and danger signs) and evaluates the added utility of including pulse oximetry and host biomarker testing for referral decision-making.”

We have also made this clear in the Methods section:

“Previous analyses reported the prognostic performance of individual host biomarkers. In the analyses described herein, data from six sites were used to derive the clinical prediction models, with the site in Cambodia prespecified a priori for held-out external geographic validation.”

Figure 1 text within this figure is ineligible to this reviewer. Please increase figure dpi.

Thank you for highlighting this. We have improved the resolution.

Clinical utility (Results): What is the confidence intervals for the estimated percentages of discharge, monitor, and referral according to each model. It is unclear to this reviewer if these differences are numerically different across models (as presented) or robust statistical measures have been used to distinguish performance of models.

We appreciate these comments and in response have now conducted additional analyses to indicate the statistical uncertainty of the results.

The projected triage dispositions (refer, monitor, and discharge) and the number needed to test (NNT) were derived from a deterministic impact analysis applied to a hypothetical population of 10,000 children, using fixed point estimates of sensitivity, specificity, and disease prevalence derived from the validation cohort. These projections are intended to illustrate the potential impact under fixed assumptions and are not inferential estimates. Consequently, confidence intervals were not calculated for the projected triage dispositions or the NNT.

In response to your suggestions, for inferential comparisons of model performance, we have now compared sensitivity, specificity, NLR, and PLR across models at the prespecified rule-in and rule-out thresholds. Differences have been estimated with 95% confidence intervals derived using bootstrap resampling procedures. Uncertainty is therefore quantified at the level of the estimated diagnostic performance metrics; the downstream impact projections follow deterministically from these parameters. These results can be found following the *Clinical Utility* section of the Results. For completeness, we have also presented results of all statistical comparisons in the appendix (Tables S7; p10). We have also described these procedures in the statistical methods section.

Clinical utility estimates inform number needed to treat analyses. Again, whether these are statistically different across models is not clear to this reviewer.

Thank you. Please see our previous response. We hope it is now clear that:

- The *pulse oximetry model* has statistically superior rule-out (sensitivity) and rule-in (specificity) performance compared to the *clinical model* and WHO danger signs;
- The *sTREM1 model* has statistically superior rule-out (sensitivity) and rule-in (specificity) performance compared to the WHO danger signs, and statistically superior rule-out (sensitivity) performance compared to the *clinical model*;
- The *clinical model* has statistically superior rule-in (specificity) performance and borderline statistically superior rule-out (sensitivity) performance compared to the WHO danger signs.

To limit the number of comparisons, we have not conducted comparisons with the *combined model* as the performance measures are very similar and it is therefore unlikely to be the preferred choice compared to either the *pulse oximetry* or *sTREM1 models*.

Cost-effectiveness models can be better described in the manuscript and supplement. How were cost of monitoring patients accounted for in estimates?

Thank you for raising this. Monitored patients were assumed to incur the cost of an additional outpatient appointment. We have now added additional details to explain how the costs of assessing, monitoring, referring, and treating patients were accounted for in the three different

dispositions (discharge, monitor, and refer) as a footnote under the parameter estimate table in the appendix:

“All patients were assumed to have an outpatient appointment cost for their initial assessment. Discharged patients had no additional costs. Referred patients with a non-severe outcome incurred a referral cost and an inpatient cost (daily cost multiplied by length of stay). Referred patients with a severe outcome incurred a referral cost and a vital organ support cost per patient (entire stay). Monitored patients with a non-severe outcome incurred an additional outpatient appointment cost. Monitored patients with a severe outcome incurred an additional outpatient appointment cost, referral cost, and a vital organ support cost per patient (entire stay).”

We have also added an additional limitation in the Discussion to highlight some of the assumptions underlying the cost effectiveness analyses and the need for further work:

“Our results indicate that incorporating either pulse oximetry or host biomarker testing are likely to be cost effective approaches to triage, however the modelling outputs are inherently limited by assumptions, for example that children recommended for monitoring and who develop severe disease can be recognised and escalated quickly, and assessing real world impact will require interventional trials.”

The discussion section can be made more succinct.

We apologise if you found the Discussion section long-winded. We have re-read the manuscript and are unable to identify portions of the Discussion that can be removed without losing important meaning. If there are specific sections that you or the Editors feel can be removed or shortened we would be happy to review these.

In several instances, results are presented as though they are reporting results from prospective testing of models. The manuscript would benefit from adjusting language to indicate that what is being presented is predicted effect of risk-models if implemented in the real-world.

Thank you for raising this important point. That was not our intention and we have moderated the language throughout to ensure this is now clear. We also hope that separation of the inferential analyses (clinical utility) and projected deterministic impact analysis will also help in this regard.

Reviewer #2

The authors of the study developed and validated a clinical prediction model to guide referral of febrile children in resource limited settings. I would like to congratulate the authors for this extraordinary study which could significantly impact the health outcomes of children with febrile illness in low-income countries. Overall, the manuscript is well written and methodologically sound.

Thank you for this very positive review, which we sincerely appreciate.

My main criticism is centred around the deviations from the original protocol and lack of clarity of selection of the validation cohort.

We agree with this criticism and have provided specific responses in relation to these aspects in the points below.

Need to cite the reference of the published study protocol

We have now referenced this at the very start of the Methods and also in the new supplementary text provided to explain the sample size calculations.

Was Cambodia selected as validation cohort a priori? If that is the case the published protocol does not indicate that. The protocol indicates "a validation cohort will be geographically distinct". It is unclear what geographically distinct cohort means.

Yes, Cambodia was selected a priori. We have now clarified this in the manuscript. The selection was made whilst the study was still recruiting and before any data were analysed. At the time the protocol was published, the Covid-19 pandemic had just been announced and so whilst it was clear the validation dataset would be geographically distinct (i.e. one country's dataset would be held-out for external validation), it was uncertain which site this would be due to the challenges to recruitment posed by the pandemic.

The original sample size for derivative and validation cohort indicated 280 and 100 outcomes,

respectively. Authors need to explain how their final sample size could affect their accuracy.

Thank you for highlighting this. Our aim was to provide a concise description of the sample size calculation in the paper but agree that this may confuse readers if read in conjunction with the original published protocol. In response to your query, we have now explained why the sample size calculation differed from the original protocol in the appendix (p20) and highlighted the low number of events in the validation dataset as a specific limitation in the Discussion:

“Nonetheless, further external validation is needed, particularly given the relatively few events in the validation dataset (n=36), which contributed to imprecision in the effect estimates. We have published our full models to encourage independent validation.”

Sample size calculations for clinical prediction model development depend, in part, on outcome prevalence, with more events required in high prevalence settings to avoid overfitting.⁷ An innovative aspect of our study was the use of stratified recruitment to enable development of a clinical prediction model in a low prevalence setting.⁸ Stratifying recruitment by admission status allowed enrolment of a random selection of outpatients, which were then appropriately weighted in the analyses. This enabled us to ensure an analysis population that approximated community care settings in terms of participant characteristics and outcome prevalence.

In the published protocol, we incorrectly used the raw (unweighted) anticipated outcome prevalence (13%) in the original sample size calculation, which led to an estimate of 14 events per parameter for derivation of the prediction models to minimise overfitting. This error was recognised whilst the study was still recruiting. At this point, and before any data were analysed, we recalculated the sample size using the correct (weighted) anticipated outcome prevalence of 1%. This resulted in an estimate of six events per parameter. Accordingly, as there were 97 events in the derivation dataset, a maximum of 16 parameters could be included for development of the prediction models. The *combined model* (including clinical parameters, oxygen saturation, and one host biomarker) comprised the largest number of candidate predictors (13 parameters), which remained within the limit estimated in the revised sample size calculations. Hence, we are confident that the derivation dataset was adequately powered for development of all prediction models.

Although there were sufficient events for robust model derivation, the validation dataset contained 36 events and, notwithstanding that sample size calculations for external validation are an area of active research (the 100 events referenced in the original protocol being an approximate ‘rule of

thumb'),⁹ we acknowledge that model validation was likely underpowered. As explained in the manuscript, this was largely a result of the unprecedented circumstances related to the Covid-19 pandemic, which resulted in delayed and interrupted recruitment, reducing the number of outcome events that we were able to accrue before recruitment had to close, in order that the study remained within the funding envelope. We note however that although the confidence intervals around the diagnostic accuracy metrics (sensitivity, specificity, etc.) are relatively wide due to the smaller number of events in the validation dataset, we were still able to detect clinically and statistically significant differences between the different triage strategies.

We prioritised ensuring a robust sample size for model derivation, which usually requires a larger sample size compared to validation, accepting that the validation may be relatively underpowered. We aim to mitigate this issue by providing our model equations, in the hope that further independent external validation will be possible by other groups.

Validation cohort had higher % of chikungunya. Could this affect the prediction model?

Although we cannot be definitive, this is very unlikely to have affected model performance as none of the patients with confirmed chikungunya infection had a severe outcome.

I am a bit surprised to see lower percentage of no focus febrile illness in these countries which are frequent due to dengue/chikungunya/scrub typhus etc. Is there possibility of selection bias in their enrolment process?

We do not think there was any major selection bias in our enrolment processes. Inpatients were recruited consecutively and a random selection of outpatients were recruited determined by off-site computer-generated random number tables. We agree that a number of pathogens that cause acute infections in Asia are synonymous with undifferentiated febrile illnesses, but note that these are often more prevalent in older children and adults and are perhaps overrepresented by research studies focussed on these ages and that test for these organisms, often in preference to “common” or “garden” pathogens. In young children, as per the rest of the world, viral respiratory tract infections and diarrhoeal diseases dominate, and this is reflected in our cohort.

Could authors add organ support breakdown (number of children receiving ventilation, renal replacement etc.). This is of interest if this is mainly driven by respiratory syndrome considering the % of severe cases with LRTI/URTI and better performance of pulse oximetry model.

Thank you for this suggestion. We have added the breakdown of organ support to the results: among the 111 survivors who required organ support 43 received mechanical ventilation, 59 received non-invasive ventilation, and 19 received inotropic therapy. We agree that the high proportion of respiratory presentations and outcome events due to respiratory-related deteriorations may partly explain the benefits of pulse oximetry in this cohort – but note that this spectrum of illness is reflective of the prevalent causes of common childhood infections. We have added a statement in the Discussion to reflect this point:

“Respiratory presentations were highly represented in our cohort and a majority of the children who developed severe disease required respiratory support. This may partly explain the observed benefits of pulse oximetry but nonetheless reflects the prevalent spectrum of common childhood infections.”

In limitations authors should discuss the deviations from protocol and how that impact their results.

We hope our above response has clarified that the deviation from the protocol with regards the sample size for the derivation dataset arose due to a mistake in the original protocol. We have now clarified this in the appendix (p20) and explained that we are confident that the model development process was adequately powered. The deviation from the protocol with regards the sample size for the validation dataset arose due to the challenges posed by the Covid-19 pandemic, which limited recruitment rate and duration. The main impact of this was fewer outcome events in the validation dataset, which contributed to imprecise effect estimates. This has been added as a specific limitation in the Discussion:

“Nonetheless, further external validation is needed, particularly given the relatively few events in the validation dataset (n=36), which contributed to imprecision in the effect estimates. We have published our full models to encourage independent validation.”

Please add the admission type distribution of “outpatient” and “inpatient” status at the time of enrolment

We have added this to Table 1.

The presenting syndrome percentage adds to >100. Need to indicate if they are not mutually exclusive

Presenting syndromes were not mutually exclusive. We have clarified this in the footnote of Table 1.

Figure 3: For category ii/iii, should the referred patients be “RED”? Amber color is both for discharged and referred group. Need clarification.

Thank you for highlighting that this was not clear. The colour coding in this figure relates to triage *appropriateness* rather than the specific triage category. If participants were triaged to the “correct” group (i.e. Category I [non-severe] to discharge; Category II/III [moderately severe] to monitor; and Category IV [severe] to refer) then they are coloured green in the bar plot. If participants are triaged to a group adjacent to their “correct” group (i.e. Category I [non-severe] to monitor; Category II/III [moderately severe] to discharge or refer; and Category IV [severe] to monitor) then they are coloured amber in the bar plot. If participants are triaged to a group that is non-adjacent (further away) from their “correct” group (i.e. Category I [non-severe] to refer; and Category IV [severe] to discharge) then they are coloured red in the bar plot. We have clarified this in the legend of the Figure.

Reviewer #3

The study titled “Referral of febrile children in resource-constrained community settings in Asia (Spot Sepsis) – a multi-country, prospective cohort study” by Chandna et al. investigates alternative approaches to improve the identification of febrile children < 5 years of age who are at risk of developing severe disease in a multi-centre, multi-country study design. The modelling approach is based on an impressive dataset and outputs suggested that adding pulse oximetry and/or sTREM 1 to management guidelines could outperform current WHO danger-sign criteria in predicting severe disease.

Thank you for this positive feedback.

The modelling strategy is innovative and represents an important step toward evidence-based refinement of referral guidelines; however, the population-level impact appears modest and the authors’ interpretation of the clinical significance may be partially overstated given the small absolute effect and the need for validation in clinical trials.

We appreciate your recognition of our innovative approach and intention to inform the development of evidence-based referral guidelines. In response to your comments and those of a previous reviewer, we have moderated the language of the Results and Discussion to clarify that these are projected benefits that may be realised if the models were applied in practice. We have also more clearly separated the inferential estimates from the projected benefits estimated by the deterministic impact analyses.

Whilst we agree that the absolute improvement in recognition of children at risk of severe febrile illness (rule-out performance) is modest, this is partly a function of the use case in a very low prevalence setting (0.30%). Nevertheless, we believe that there could be substantial public health impact. As we note in the Discussion, these are conservative estimates, focussed on identification of the most severely ill children. It is probable that a substantially larger proportion of children would merit hospital referral. Our secondary analyses utilising the categorical outcome indicate that the models provide discrimination across the severity spectrum and thus further gains can be anticipated. Furthermore, an equally important component is the very large reduction in unnecessary referrals (improved rule-in performance) that could be realised by model-based triage compared to WHO danger signs. In remote and conflict-affected settings, referrals carry substantial opportunity costs and better targeting of referrals could have considerable benefit for patients and health systems.

Overall, the study shows exceptional quality and provides a constructive discussion of its limitations that may inspire further research in this important field. The used methodology seems appropriate and thorough. However, I suggest seeking further expertise from additional reviewers regarding the development of the model as my experience is limited in this regard.

Thank you for this very complimentary feedback.

Statistics provided are appropriate, transparent and thoroughly reported.

We appreciate this.

As noted above, the overall impact of the study appears overemphasised given the comparatively small absolute effect at the population level. This limitation could be more clearly acknowledged throughout the manuscript. Nevertheless, the findings are robust, methodologically sound, and have the potential to inform global guideline development, warranting further evaluation in clinical trials.

Please see our previous response and moderation of the language throughout the Results and Discussion to strike a more cautious tone and reflect the projected nature of the impact analysis. We hope that separating the clinical utility and impact analysis will also help in this regard.

Line 194-210: This section is critical and summarises the main study findings well while acknowledging limitations of number of outcome events. However, the article generally appears to overstate the population-level impact of the models to my understanding. Although every child's life is invaluable, the authors should emphasise that the absolute effect remains small, with high NNT, even if the models offer improved discrimination and could serve as an adjunct to existing WHO guidelines following clinical evaluation in clinical trials.

Thank you for these comments. In response to your comments and those of a previous reviewer, we have restructured this section of the manuscript to separate the clinical utility and projected impact analysis. With regards the potential population-level impact, please see our previous response which outlines the conservative nature of the estimate related to improved rule-out performance (NNT)

which focuses on the most severely ill children, as well as the importance of the considerable improvement in rule-in performance, in terms of better targeting of referrals in locations where they can be very costly.

Line 56-58: As an economic evaluation is added in this study, authors could consider adding the financial barrier as additional reason of delayed care.

Thank you for this suggestion. We have added this.

Line 82: Authors should consider to omit this sentence to avoid confusion and discuss this aspect in the discussion section in respect to the comments below.

Thank you. We have removed this sentence.

Line 88: The authors should consider to explicitly state that the prognostic models were developed using the same patient cohort in which the biomarkers' performance was previously evaluated, to ensure transparency and prevent overinterpretation. This could be added to the methods section.

Thank you. In response to your comments and those a previous reviewer we have now explicitly stated that this work uses the same dataset as our previous study:

“Building on our previous work and using the same multi-country dataset, our objective in this manuscript was to address a different, complementary question: how prognostic information can be integrated into risk prediction models to guide referral decisions for febrile children in resource-constrained community contexts and decentralised models of care. Whereas our previous analysis focused on the predictive performance of individual host biomarkers for disease progression, the present study develops and validates multivariable clinical prediction models comprising simple clinical parameters (for example, vital and danger signs) and evaluates the added utility of including pulse oximetry and host biomarker testing for referral decision-making.”

We have also added a sentence in the Methods to this regard:

“Previous analyses reported the prognostic performance of individual host biomarkers. In the analyses described herein, data from six sites were used to derive the clinical prediction models, with the site in Cambodia prespecified a priori for held-out external geographic validation.”

Line 177: Add n/N (25%)

Thank you for this suggestion. This percentage (25%; the proportion of children who progress to severe disease who would have been missed) is the inverse of the sensitivity (74.7%) of the model. For this example, the n/N would be 9/36. However, we report a number of similar percentages in this paragraph for the sensitivity and specificity of the various models at the prespecified decision thresholds. Due to the weighted analyses, it is not possible to provide a simple fraction for the specificities. Therefore, for consistency and to improve readability of the paragraph, we have opted not to include the n/N. We hope you agree this is appropriate.

Line 277-281: Authors should review this suggestion as availability of oxygen supply may be limited in primary healthcare settings and rural clinics in the respective areas. This shouldn't downplay the importance of pulse oximetry; however, it should be taken in account that experience handling hypoxaemia may and probably will further be limited.

Thank you for this comment. Our intention was to convey the concept that in better-resourced primary care settings (in terms of equipment, supply chains, and health worker capacity), pulse oximetry may be preferred to host biomarker testing, given the improvements in both sensitivity and specificity over the *clinical model* (vs. only improvements in sensitivity of the *sTREM1 model*). However, in more peripheral settings (for example, community health workers), host biomarker testing may be preferable, given the improvement in sensitivity over the *clinical model* (and sensitivity and specificity compared to WHO danger signs), the fact that universal rapid diagnostic testing for malaria is often performed, and acknowledging that pulse oximetry may not be feasible in these contexts. We do recognise your point that pulse oximetry may also not be feasible at many primary health centres and rural clinics and have now moderated the language to reflect this:

“The pulse oximetry model outperformed the sTREM1 model and may be the preferred option in many settings. Its adoption would be particularly suited to primary health centres and rural

clinics that have access to safe storage, secure supply chains, supplemental oxygen therapy, and staff trained to manage other conditions for which a pulse oximeter may provide benefit.”

Line 281-284: Authors should review this recommendation for two reasons. Overinterpretation of malaria RDTs is a growing concern globally, particularly in sub-Saharan Africa, where over-prescription of antimalarials is driving resistance, and leads to underestimation of other diseases with potentially underlying chronic malaria infection. As no clear cut-off for the proposed biomarker is available and its clinical relevance is yet to be assessed, authors should consider softening the wording of their recommendation; otherwise introducing the proposed assays could lead to overinterpretation of “positive” tests. However, while a positive malaria RDT suggests direct action (> antimalarials), incorporating host biomarkers into a (combined) RDT for community health workers is simplifying complex clinical decisions which is out of scope for this profession (Is initiating antimalarial treatment sufficient to prevent progression to severe disease? Does a positive sTREM-1 always indicate hospital admission? If yes, which level of care should be considered as severe disease may be more likely?).

Thank you for these insightful comments. We agree that management of febrile illness in resource-limited community settings is challenging, in part due to lack of diagnostic tests. Whilst coincident (asymptomatic) malaria infection in the context of another cause of febrile illness poses a challenge in malaria endemic regions of sub-Saharan Africa, this is not typical in Asia.^{10,11} Indeed, maintaining malaria testing is critical to support regional elimination efforts and expanding primary care provision beyond malaria testing and treatment sustains attendance of febrile patients.¹² A pressing concern is the realisation of the anticipated overprescription of antibiotics in patients who test negative for malaria, driven, in part, because community healthcare providers lack tools to distinguish non-malarial infections that require antibiotics from those that are self-limiting.¹³ Improving this situation is a primary focus of our research group.¹⁴ Previous work indicates that identifying patients at risk of disease progression is an essential component of improved community-based fever management strategies,¹⁵ and that introduction of multiplexed diagnostic tests may be feasible and economically advantageous.^{16,17} Of note, we would not necessarily advocate for a specific sTREM1 (or SpO₂) cut-off. Rather, in preference to dichotomising these variables they could be incorporated into multivariable risk prediction models. These could be housed within a smartphone-based application to support community healthcare providers in their assessments of febrile patients.¹⁸⁻²⁰

Authors could consider discussing the performance of sTREM 1 regarding their study cohort given the distribution of pathogens. The study team identified very few bacterial infections where sTREM 1 was found to be particularly valuable in identifying severe disease to my knowledge.

Thank you for this suggestion. We agree that sTREM1 has been found to be valuable in bacterial infections.^{21,22} It has also been shown to be useful in many other contexts, including malaria and childhood pneumonia, which is predominantly viral in aetiology.^{2-4,23} The underlying hypothesis of our research is that host biomarkers of final common pathways are required for risk stratification at the primary care level, where the cause of infection is usually unknown at the time of presentation.

The choice and use of references are thorough and appropriate for the purpose of the manuscript.

Thank you.

In respect to the above-mentioned aspects, abstract, introduction, and conclusions are coherent and appropriately aligned with the study objectives. The provided background is adequate enabling readers to understand the rationale and significance of the work.

Thank you.

REFERENCES

1. Chandna A, Koshiaris C, Mahajan R, et al. Risk stratification of childhood infection using host markers of immune and endothelial activation in Asia (Spot Sepsis): a multi-country, prospective, cohort study. *Lancet Child Adolesc Health* 2025; **9**(9): 634-45.
2. Leligdowicz A, Conroy AL, Hawkes M, et al. Risk-stratification of febrile African children at risk of sepsis using sTREM-1 as basis for a rapid triage test. *Nat Commun* 2021; **12**(1): 6832.
3. Balanza N, Erice C, Ngai M, et al. Prognostic accuracy of biomarkers of immune and endothelial activation in Mozambican children hospitalized with pneumonia. *PLOS Glob Pub Health* 2023; **3**(2): e0001553.
4. Jullien S, Richard-Greenblatt M, Ngai M, et al. Performance of host-response biomarkers to risk-stratify children with pneumonia in Bhutan. *J Infect* 2022; **85**(6): 634-43.
5. de Hond AAH, Steyerberg EW, van Calster B. Interpreting area under the receiver operating characteristic curve. *Lancet Digit Health* 2022; **4**(12): e853-e5.
6. Collins GS, Reitsma JB, Altman DG, Moons KG. Transparent Reporting of a multivariable prediction model for Individual Prognosis or Diagnosis (TRIPOD): the TRIPOD statement. *Ann Intern Med* 2015; **162**(1): 55-63.
7. Riley RD, Snell KI, Ensor J, et al. Minimum sample size for developing a multivariable prediction model: PART II - binary and time-to-event outcomes. *Stat Med* 2019; **38**(7): 1276-96.
8. Holtman GA, Berger MY, Burger H, et al. Development of practical recommendations for diagnostic accuracy studies in low-prevalence situations. *J Clin Epidemiol* 2019; **114**: 38-48.
9. Collins GS, Ogundimu EO, Altman DG. Sample size considerations for the external validation of a multivariable prognostic model: a resampling study. *Stat Med* 2016; **35**(2): 214-26.
10. Asmelash D, Agegnehu W, Fenta W, Asmelash Y, Debebe S, Asres A. The Burden of Asymptomatic Malaria Infection in Children in Sub-Saharan Africa: A Systematic Review and Meta-Analysis Exploring Barriers to Elimination and Prevention. *J Epidemiol Glob Health* 2025; **15**(1): 17.
11. Kotepui M, Kotepui KU, Masangkay FR, Mahittikorn A, Wilairatana P. Prevalence and proportion estimate of asymptomatic Plasmodium infection in Asia: a systematic review and meta-analysis. *Sci Rep* 2023; **13**(1): 10379.
12. McLean ARD, Wai HP, Thu AM, et al. Malaria elimination in remote communities requires integration of malaria control activities into general health care: an observational study and interrupted time series analysis in Myanmar. *BMC Med* 2018; **16**(1): 183.
13. Hopkins H, Bruxvoort KJ, Cairns ME, et al. Impact of introduction of rapid diagnostic tests for malaria on antibiotic prescribing: analysis of observational and randomised studies in public and private healthcare settings. *BMJ* 2017; **356**: j1054.
14. Chandna A, Shwe Nwe Htun N, Peto TJ, et al. Defining the burden of febrile illness in rural South and Southeast Asia: an open letter to announce the launch of the Rural Febrile Illness project. *Wellcome Open Research* 2021; **6**: 64.
15. Chandna A, Osborn J, Bassat Q, et al. Anticipating the future: prognostic tools as a complementary strategy to improve care for patients with febrile illnesses in resource-limited settings. *BMJ Glob Health* 2021; **6**(7): e006057.
16. Visser MT, Lek D, Adhikari B, et al. Operational evaluation of the deployment of Malaria/CRP Duo and Dengue Duo rapid diagnostic tests for the management of febrile illness by village malaria workers in rural Cambodia. *BMC Infect Dis* 2025; **25**(1): 679.
17. Lubell Y, Chandna A, Smithuis F, et al. Economic considerations support C-reactive protein testing alongside malaria rapid diagnostic tests to guide antimicrobial therapy for patients with febrile illness in settings with low malaria endemicity. *Malar J* 2019; **18**(1): 442.
18. Tan R, Kavishe G, Luwanda LB, et al. A digital health algorithm to guide antibiotic prescription in pediatric outpatient care: a cluster randomized controlled trial. *Nat Med* 2024; **30**(1): 76-84.
19. Keitel K, Kagoro F, Samaka J, et al. A novel electronic algorithm using host biomarker point-of-care tests for the management of febrile illnesses in Tanzanian children (e-POCT): A randomized, controlled non-inferiority trial. *PLoS Med* 2017; **14**(10): e1002411.
20. Chew R, Wynberg E, Liverani M, et al. Evaluation of an electronic clinical decision support algorithm to improve primary care management of acute febrile illness in rural Cambodia: protocol for a cluster-randomised trial. *BMJ Open* 2024; **14**(10): e089616.
21. Richard-Greenblatt M, Boillat-Blanco N, Zhong K, et al. Prognostic Accuracy of Soluble Triggering Receptor Expressed on Myeloid Cells (sTREM-1)-based Algorithms in Febrile Adults Presenting to Tanzanian Outpatient Clinics. *Clin Infect Dis* 2020; **70**(7): 1304-12.

22. Chen HL, Hung CH, Tseng HI, Yang RC. Soluble form of triggering receptor expressed on myeloid cells-1 (sTREM-1) as a diagnostic marker of serious bacterial infection in febrile infants less than three months of age. *Jpn J Infect Dis* 2008; **61**(1): 31-5.
23. O'Brien KL, Baggett HC, Brooks WA, et al. Causes of severe pneumonia requiring hospital admission in children without HIV infection from Africa and Asia: the PERCH multi-country case-control study. *Lancet* 2019; **394**(10200): 757-79.

Editorial comments

Per Nature Medicine citation style, please place the citation number before the punctuation, not after.

We have adjusted the citation formatting as requested.

You currently have 7 main display items, and 2 supplementary figures. Please convert 1 of your main display items (I suggest Figure 4) to extended display figures to enhance visibility (each of which must fit on a single page). Please note that we can permit 6 main display items and 10 Extended Data display items. Please convert all your supplementary figures into extended display items.

We have converted Figure 4 to an Extended Data Display Item. We have also converted our two supplementary figures to Extended Data Display Items.

Please be aware that we are unable to accommodate supplementary text. Please integrate them into either the main text or the methods section. Please note that there is no limit for our Methods section as it is online only.

We have integrated all supplementary text into the main manuscript. The supplementary material now only contains Tables.

The supplementary material cannot include references. Please move the references in the supplementary material to the main reference list and ensure that this is appropriately cited in the main text.

We have removed all references from the supplementary material and moved these to the main manuscript.

Please format your Spot Sepsis Investigator Group. For consortium authorship, we do not allow tables. Please format your authors and affiliations for this consortium similar to the title page.

We have adjusted the formatting of the consortium authorship to follow a similar style to the title page.

Please list the Ethics approvals in the Methods section, rather than having it as a supplementary table.

We have removed this supplementary table and incorporated the ethics approvals into the Methods section of the main manuscript.

In the data availability statement, please provide an estimated timeline between submission and application to request data access and decision.

We have indicated that the timeframe between submission request and decision is anticipated to be approximately one month.

Please provide a code availability statement. We do not allow “code available upon request”. Please either deposit the code in a repository, such as GitHub and provide the appropriate details. Alternatively, please provide more information for readers on any restrictions to accessing the code and how the code can be accessed, including how the data can be accessed upon application, the contact details and timelines.

We have included a code availability statement and indicated that the timeframe between submission request and decision is anticipated to be approximately one month.

Please remove the “Open Access” section

We have retained the Open Access statement, as this is a requirement of our funder (Wellcome): <https://wellcome.org/research-funding/guidance/open-access-guidance/complying-with-our-open-access-policy>. We hope this is acceptable.

Any references cited only in the methods needs to be included in a separate methods-only references section and should be numbered contiguously to the main reference list (i.e. number starts at XX following on from the numbering of the main reference list, not 1).

We have implemented this and created two separate reference lists with contiguous numbering between them.

Please include an "Ethics and Inclusion statement" in the methods section.

We have included this as a separate statement to the Ethics approvals.

The article file must only contain these items in this order:

- Title

- Author List and affiliations
- Abstract
- Introduction
- Results (with Subheadings)
- Discussion
- Acknowledgements
- Author Contributions
- Competing Interests Statement
- References (for main text only)
- Figure legends (for main text only)
- Tables (note: tables should be pasted into Word files as editable tables, not as images)
- Methods
- Data Availability Statement
- Code Availability Statement
- Methods-only References

We have followed this format for the main article file.

In addition to addressing the remaining points from the reviewers, please edit your manuscript to comply with our formatting guidelines for Articles, which are:

* Abstract: 200 words, unreferenced.

The abstract is unreferenced and the word count is 195 words.

* Main text: 4000 words with subheadings for the Introduction, Results and Discussion

The word count for the main text is 3,777 words.

* References: up to 60 in the main text + 20 methods-only references

The main text contains 58 references. As a result of moving all the references from the supplementary material there are 34 references in the methods-only references section. We hope this is acceptable.

* Display items: up to 6 main display items (inclusive of figures and tables) and up to 10 Extended Data display items (inclusive of figures and tables). Extended Data are an integral

part of the paper and only data that directly contribute to the main message should be presented.

We include 6 main display items (3 tables and 3 figures) and 2 extended data display items (3 figures).

* Online Methods: no word limit; please provide the methods consolidated in a single section at the end of the main text document

We include the Methods section at the end of the main text document.

Reviewer comment

All reviewer comments have been adequately addressed. I congratulate the authors on their impressive work and support publication of the manuscript in its current form. I have only one minor suggestion regarding wording in the abstract. The term “simple clinical parameters” is potentially misleading and should be reconsidered. I suggest omitting the sentence and revise the following one to: “Including either pulse oximetry or the host biomarker sTREM-1 substantially improved identification of children at risk of severe disease (death or organ support within two days) and outperformed WHO danger signs while increasing sensitivity [...]”

We are glad to have been able to address your previous comments and thank you for this suggestion. We apologise that the Abstract was not clear. We do think it is important to emphasise that the basic clinical model outperformed the WHO danger signs (current standard of care). The opportunity cost of including either pulse oximetry or host biomarker testing is greater than using clinical parameters alone and there may be certain resource-limited primary care settings where these measurements are not feasible. In response to your comment, to make clear the improved performance achieved by the basic clinical model, we have included the sensitivity and specificity for both the clinical model and the WHO danger signs in the Abstract:

“The model using simple clinical parameters (sensitivity 74.7% [95% CI 59.4 to 88.1]; specificity 99.1% [97.7 to 99.7]) outperformed WHO criteria (sensitivity 55.5% [95% CI 39.4 to 72.7]; specificity 82.6% [95% CI 77.1 to 87.6]).”